

# Impact of halogen chemistry on air quality in coastal and continental Europe: application of CMAQ model and implication for regulation

Qinyi Li[1], Rafael Borge[2], Golam Sarwar[3], David de la Paz[2], Brett Gantt[4], Jessica Domingo[2], Carlos A. Cuevas[1], and Alfonso Saiz-Lopez[1]*

1 Department of Atmospheric Chemistry and Climate, Institute of Physical Chemistry Rocasolano, CSIC, Madrid 28006, Spain
2 Environmental Modelling Laboratory, Department of Chemical & Environmental Engineering, Universidad Politécnica de Madrid (UPM), Madrid, Spain
3 National Exposure Research Laboratory, Environmental Protection Agency, Research Triangle Park, NC 27711, United States
4 Office of Air Quality Planning and Standards, Environmental Protection Agency, Research Triangle Park, NC 27711, United States

*Correspondence to: Alfonso Saiz-Lopez (a.saiz@csic.es)

**Abstract:**

Halogen (Cl, Br, and I) chemistry has been reported to influence the formation of secondary air pollutants. Previous studies mostly focused on the impact of chlorine species on air quality over large spatial scales. Very little attention has been paid to the effect of the combined halogen chemistry on air quality over Europe and its implications for control policy. In the present study, we utilize a widely-used regional model, the Community Multiscale Air Quality Modeling System (CMAQ), incorporated with the latest halogen sources and chemistry, to simulate the abundance of halogen species over Europe and to examine the role of halogens in the formation of secondary air pollution. The results suggest that the CMAQ model is able to reproduce the level of $O_3$, $NO_2$, and halogen species over Europe. Chlorine chemistry slightly increases the levels of OH, $HO_2$, $NO_3$, $O_3$, and $NO_2$ and substantially enhances the level of Cl radical. Combined halogen chemistry reduces the $HO_2$/OH ratio by decreasing the level of $HO_2$ and increasing OH, significantly reduces the concentrations of $NO_3$ and $O_3$, and decreases $NO_2$ in the highly polluted regions and increases $NO_2$ in other areas. The maximum effects of halogen chemistry occur over oceanic and coastal regions, but some noticeable impacts also occur over continental Europe. Halogen chemistry affects the number of days exceeding the European Union target threshold for the protection of human being and vegetation from ambient $O_3$. In light of the significant impact of halogen chemistry on air quality, we recommend that halogen chemistry be considered for inclusion in air quality policy assessments, particularly in coastal cities.


## 1. Introduction

Halogen (Cl, Br, and I) species and related processes have been known to deplete stratospheric

ozone ($O_3$) for several decades (Molina and Rowland, 1974; Farman et al., 1985). In the

troposphere, it has only been recognized recently that halogen species affect the concentration of

air pollutants, e.g., directly destroying $O_3$ (R1), influencing the $NO/NO_2$ ratio (R2) and the

$HO_2/OH$ ratio (R3 and R4) (Saiz-Lopez and von Glasow, 2012; Simpson et al., 2015). The

budgets of $NO_x$ ($NO+NO_2$) and $HO_x$ ($OH+HO_2$) also affect the formation of $O_3$ (e.g., Sillman,

1999; Li et al., 2018).

$$X (Cl, Br, I) + O_3 \rightarrow XO + O_2 \tag{R1}$$

$$XO + NO \rightarrow X + NO_2 \tag{R2}$$

$$XO + HO_2 \rightarrow HOX + O_2 \tag{R3}$$


$$HOX + hv \rightarrow OH + X \tag{R4}$$

Chlorine radical (Cl) initiates the oxidation of hydrocarbons (methane, $CH_4$, and non-methane

volatile organic compounds, NMVOC, R5) in a similar way to OH radical, reducing the lifetime

of $CH_4$ and NMVOC and leading to the formation of $O_3$ in the presence of $NO_x$ (Thornton et al.,

2010).


$$RH + Cl \rightarrow HCl + RO_2 \tag{R5}$$

The combined effect of halogen chemistry on air quality, therefore, is complicated and depends

heavily on local conditions, e.g., atmospheric compositions, oxidative capacity, etc. (Sherwen et

al., 2016; Muñiz-Unamunzaga et al., 2018). Evaluation of the complex role of halogen

chemistry in air quality requires the employment of advanced, high-resolution chemical

transport models.

A number of modeling studies have been conducted to investigate the impact of individual

halogen species on air quality. The chemistry of chlorine, mainly that of $ClNO_2$, has been

reported to increase the oxidation capacity and the formation of $O_3$ in recent studies (Sarwar et

al., 2012, 2014; Li et al., 2016). Bromine and iodine (Br and I) chemistry are reported to

decrease the concentration of $O_3$ over the oceanic and terrestrial regions (Fernandez et al., 2014;

Saiz-Lopez et al., 2014).



Only a few regional modeling studies have explored the combined influence of the halogen chemistry on air quality. The first modeling study with combined halogen (Cl, Br, and I) chemistry was conducted by Sarwar et al. (2015) who used a hemispheric version of the Community Multiscale Air Quality (CMAQ) model (Ching and Byun, 1999; Byun and Schere, 2006; Mathur et al., 2017) to explore the effect of bromine and iodine chemistry on tropospheric $O_3$ over the Northern Hemisphere. Gantt et al. (2017) then utilized the CMAQ model to explore the role of halogen chemistry at a regional scale over the continental United States (US). While these studies focused on the hemispheric impact or over the continental US, Muñiz-Unamunzaga et al. (2018) applied the full-halogen chemistry version of CMAQ to examine the effect of the halogen sources on air quality at a city scale (4 km resolution) in Los Angeles, California, US.

The regulation of air quality and the control of air pollutants emission in Europe started in the early 1970s and over forty years of effort has successfully improved air quality throughout Europe (http://ec.europa.eu/environment/air/index_en.htm). Nonetheless, poor air quality persist in major cities like Madrid, Paris, and London (EEA, 2018a). To our best knowledge, the only modeling study including halogen chemistry in Europe was conducted by Sherwen et al. (2017) who used a global model, GEOS-Chem, in a regional configuration (with a grid size of 0.25° × 0.315°, ~25km) to examine the effect of halogens on air quality. Considering that the grid size has a noticeable impact on air quality model predictions (Gantt et al., 2017; Sherwen et al., 2017), it is important to conduct high-resolution simulations using regional models to examine the overall effect of halogen species on air pollution over Europe and to assess potential air quality policy implications.

In this study, we use a state-of-the-art regional chemical transport model (CMAQ) with 12 km horizontal resolution, instrumented with comprehensive halogen sources and chemistry (Sarwar et al., 2015), to simulate the levels of halogen species over Europe, examine the effect on the oxidation capacity and the concentration of air pollutants, and explore the potential implications for air quality policy related to $NO_2$ and $O_3$.



## 2. Method and Materials

### 2.1 Data

The meteorological inputs for the CMAQ model were obtained from the Weather Research and Forecasting model (WRF 3.7.1) (Skamarock and Klemp, 2008; Borge et al., 2008a). The WRF model was initialized from global reanalyses from the National Centers for Environmental Prediction (NCEP) Global Forecast System (GFS) with a spatial resolution of $1° × 1°$ and a temporal resolution of 6 h (available online at http://rda.ucar.edu/datasets/ds083.2/) which was

updated daily from NCEP global analyses with $0.5°$ resolution (available online at http://www.nco.ncep.noaa.gov/pmb/products/sst/). Besides, NCEP's ADP global upper-air (NCAR archive ds351.0) and global surface observations (NCAR archive ds461.0) were used to drive the simulation with a Newtonian relaxation technique in the WRF model.

        Anthropogenic emissions for the year 2016 were taken from the $0.1° × 0.1°$ gridded EMEP

inventory (EMEP/CEIP, 2014). It should be noted that no anthropogenic chlorine sources are included in our emission inventory. The temporal profiles and vertical distribution needed to resolve the emissions were those used in the EuroDelta experiment (van Loon et al., 2007). Biogenic emissions were estimated using the Model of Emissions of Gases and Aerosols from Nature (MEGANv2.10) (Guenther et al., 2012). All emissions were gridded to our model

domain, temporally allocated and chemically speciated using the Sparse Matrix Operator Kernel Emissions (SMOKE) model, version 3.6.5 (UNC, 2015; Borge et al., 2008b).

        In addition, we used measurement data of $NO_2$ and $O_3$ from 465 background stations (traffic and industrial stations are not included) across Europe from database *AirBase* (public air quality database system of the European Environment Agency, 2018) to compare the results of our

simulation with observations (Fig. 1). Among these stations, 340 are located in inland areas (223 for $NO_2$ and 315 for $O_3$), and 123 are located in the coastal areas (80 for $NO_2$ and 101 for $O_3$).



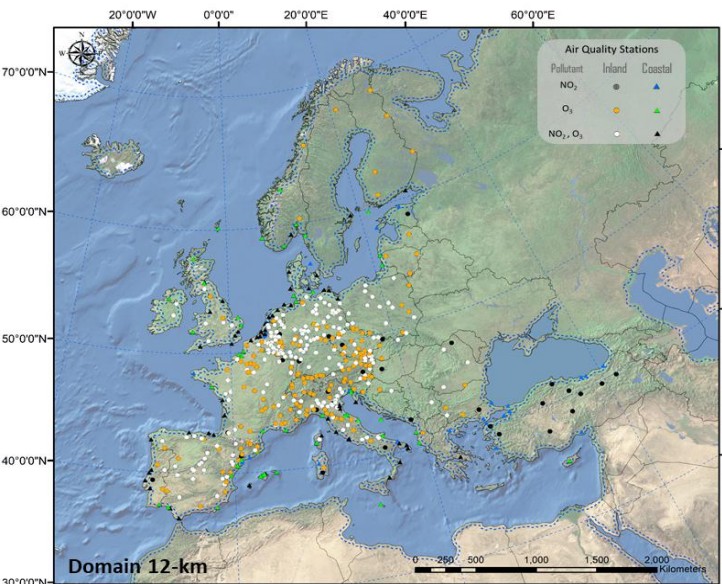

Figure 1. Geographic representation of the modeling domain and 465 air quality stations used for model evaluation

## 2.2 Modeling system

The CMAQ model is widely used and includes comprehensive representations of many essential atmospheric processes. The skill of the model in reproducing observed air quality has been demonstrated in many previous studies (Foley et al., 2010; Appel et al., 2013, 2017; Mathur et al., 2017), including applications over Europe (Borge et al., 2008a; Appel et al., 2012; Solazzo et al., 2017). CMAQ version 5.2 (www.epa.gov/cmaq; doi:10.5281/zenodo.1167892) containing the Carbon Bond chemical mechanism with halogen chemistry was used in this study (Appel et al., 2017). The chlorine chemistry includes 26 gas-phase chemical reactions (Sarwar et al., 2012). In addition, the heterogeneous hydrolysis of dinitrogen pentoxide ($N_2O_5$) can produce nitryl chloride ($ClNO_2$) and nitric acid ($HNO_3$) in the presence of particulate chloride. In the absence of particulate chloride, heterogeneous hydrolysis of $N_2O_5$ produces only $HNO_3$. The bromine chemistry contains 39 gas-phase chemical reactions and one heterogeneous reaction while the iodine chemistry contains 53 gas-phase chemical reactions (Sarwar et al., 2015).



### 2.3 Simulation setup

A detailed description of physics and other model options can be found in (de la Paz et al., 2016) (Table S1). The CMAQ modeling domain covers the entirety of Europe (Fig. 1) with 12 km horizontal resolution. The vertical extent of the model extended from the surface to 100 mbar and contained 35 layers with an average surface layer thickness of approximately 20 m. The CMAQ chemical transport model is configured to use the Piecewise Parabolic Method to

describe advection processes, the Asymmetric Convective Model (version 2) to describe vertical diffusion processes, and the multiscale method to describe horizontal diffusion processes. Gas-phase chemistry, aqueous chemistry, aerosol processes, and dry and wet deposition were also included. The Rosenbrock solver was used for gas-phase chemistry.

The study was completed for the month of July 2016 with a spin-up period of 7 days. We

performed three simulations to isolate the effect of halogen chemistry on air quality (in brackets the name of the scenario used hereafter):

(1) Base model without halogen chemistry (BASE),

(2) BASE and chlorine chemistry (CHL), and

(3) CHL and Br and I chemistry (HAL).

The BASE model simulation includes the Carbon Bond chemical mechanism but does not contain any halogen chemistry, and only the $HNO_3$ is produced from the heterogeneous hydrolysis of $N_2O_5$. The CHL simulation contains the Carbon Bond chemical mechanism with chlorine chemistry and considers $ClNO_2$ and $HNO_3$ production from the heterogeneous uptake of $N_2O_5$ on the aerosol surface. The HAL simulation contains the Carbon Bond chemical

mechanism with full halogen chemistry and produces $ClNO_2$ and $HNO_3$ from the heterogeneous uptake of $N_2O_5$ on the aerosol surface.

Boundary conditions for the model were derived from the hemispheric CMAQ simulations. Two simulations were conducted using the hemispheric CMAQ simulations: the first simulation used the Carbon Bond chemical mechanism and the chlorine chemistry, while the second simulation

used the Carbon Bond chemical mechanism and the full halogen chemistry. Results from the



hemispheric CMAQ simulation using the Carbon Bond chemical mechanism and the chlorine chemistry were used to generate boundary conditions for the BASE and CHL simulations, while results from the hemispheric CMAQ simulation using the Carbon Bond chemical mechanism and the full-halogen chemistry were used to generate boundary conditions for the HAL

simulation.

Therefore, the difference between CHL and BASE simulations represents the impact of the chlorine chemistry on air quality and the difference between HAL and BASE simulations represents the effect of halogen chemistry on air quality.

## 3. Results and Discussions

### 3.1 Evaluation of model performance

The performance of the CMAQ model in simulating air quality over Europe is evaluated using observation data collected from 465 measurement stations. We separate the stations into coastal (within 24 km from the coast) and continental stations (Fig. 1). Table 1 presents the statistics of

the model performance for $O_3$ and $NO_2$ for BASE and HAL simulations.

The BASE and HAL simulations generally reproduce the concentration levels and the temporal variations of $O_3$ and $NO_2$ both at coastal and continental stations. The correlation coefficients between simulations and observations (Fig S1 in supplement) show that CMAQ satisfactorily reproduces the variation of $O_3$ and $NO_2$ over most of Europe especially the coastal regions (> 0.7

for $O_3$ and > 0.5 for $NO_2$). The BASE simulation over-predicts $O_3$ while the HAL simulation under-predicts $O_3$ compared to observations both at coastal and continental stations (Table 1). The BASE simulation under-predicts $NO_2$ compared to observations both at coastal and continental stations (Table 1). Such an under-estimation of $NO_2$ can occur for many reasons including (1) positive artifacts of $NO_2$ monitors, (2) under-estimation of $NO_x$ in the emission

inventory, and (3) rapid transformation of $NO_2$ into $HNO_3$ in the model compared to the real atmosphere. However, model performance is reasonable as the $NO_2$ underestimation is relatively small. The HAL simulation deteriorates $NO_2$ comparison with observations by a small margin





(Table 1).

Overall, the evaluation of the CMAQ model over Europe demonstrates that the model is capable
of reproducing the levels of atmospheric chemical species and can be used to investigate the
impact of halogen chemistry on air quality over Europe. It also suggests that the incorporation of
halogen chemistry improves the model performance for $O_3$ concentrations by a small margin
while deteriorating the model performance for $NO_2$ by a smaller margin.

Table 1. Statistical summary of model performance

| Statistics | $O_3$ | | | | $NO_2$ | | | |
|---|---|---|---|---|---|---|---|---|
| | Coastal | | Inland | | Coastal | | Inland | |
| | BASE | HAL | BASE | HAL | BASE | HAL | BASE | HAL |
| MB (ppbv) | 3.2 | -2.9 | 2.9 | -1.7 | -2.5 | -2.7 | -2.7 | -2.8 |
| RMSE (ppbv) | 23.0 | 22.3 | 25.8 | 25.3 | 6.5 | 6.7 | 6.9 | 6.9 |
| r  (dimensionless) | 0.63 | 0.63 | 0.61 | 0.62 | 0.44 | 0.45 | 0.42 | 0.43 |
| IOA (dimensionless) | 0.71 | 0.72 | 0.71 | 0.71 | 0.55 | 0.55 | 0.52 | 0.52 |

Note: MB = Mean Bias, RMSE = Root Mean Square Error, r = correlation coefficient, IOA = Index of Agreement.

## 3.2 Simulated halogen species

Average surface concentrations of the inorganic halogen species predicted in the HAL
simulation over the ocean are summarized in Table 2. HCl is the dominant chlorine species with
an average level of 237.5 pptv representing over 97% to the total inorganic chlorine ($Cl_y$) while
the average $ClNO_2$ is 6.1 pptv (2.5%) and the remaining species contributing less than 1%. The
most abundant $Br_y$ species are HBr (1.9 pptv, 48.9%), HOBr (1.2 pptv, 29.8%) and $Br_2$ (0.4 pptv,
9.9%) while the remaining species contribute ~10%. HOI (7.2 pptv, 48.9%), $I_2O_3$ (4.3 pptv,
29.4%) and $IONO_2$ (1.4 pptv, 9.2%) contribute nearly 90% of $I_y$ over the ocean, while the
remaining species contribute ~10%. The predicted average concentrations of the critical halogen
radicals, Cl, BrO, and IO, are $1.0 \times 10^{-4}$ pptv, 0.2 pptv, and 0.8 pptv, respectively, over the ocean
in the Europe.

Table 2. Simulated average concentrations of inorganic halogen species over the ocean

| Species | Concentration (pptv) | Percentage (%) | Species | Concentration (pptv) | Percentage (%) | Species | Concentration (pptv) | Percentage (%) |
|---|---|---|---|---|---|---|---|---|
| HCl | 237.5 | 97.1 | HBr | 1.9 | 48.9 | HOI | 7.2 | 48.9 |
| $ClNO_2$ | 6.1 | 2.5 | HOBr | 1.2 | 29.8 | $I_2O_3$ | 4.3 | 29.4 |
| HOCl | 1.0 | 0.4 | $Br_2$ | 0.4 | 9.9 | $IONO_2$ | 1.4 | 9.2 |
| ClO | 0.1 | < 0.1 | BrO | 0.2 | 4.7 | $INO_2$ | 0.8 | 5.5 |
| $Cl_2$ | $2.0 \times 10^{-2}$ | < 0.1 | $BrONO_2$ | 0.2 | 4.5 | IO | 0.8 | 5.3 |
| Cl | $1.0 \times 10^{-4}$ | < 0.1 | $BrNO_2$ | 0.1 | 1.3 | I | 0.2 | 1.1 |
| - | | | BrCl | $2.0 \times 10^{-2}$ | 0.6 | HI | $5.0 \times 10^{-2}$ | 0.3 |
| - | | | Br | $9.0 \times 10^{-3}$ | 0.2 | $I_2$ | $4.0 \times 10^{-2}$ | 0.2 |
| - | | | - | | | INO | $1.0 \times 10^{-2}$ | 0.1 |
| - | | | - | | | $I_2O_2$ | $4.0 \times 10^{-3}$ | < 0.1 |
| - | | | - | | | $I_2O_4$ | $2.0 \times 10^{-3}$ | < 0.1 |
| Total $Cl_y$ | 244.7 | 100 | Total $Br_y$ | 3.9 | 100 | Total $I_y$ | 14.8 | 100 |

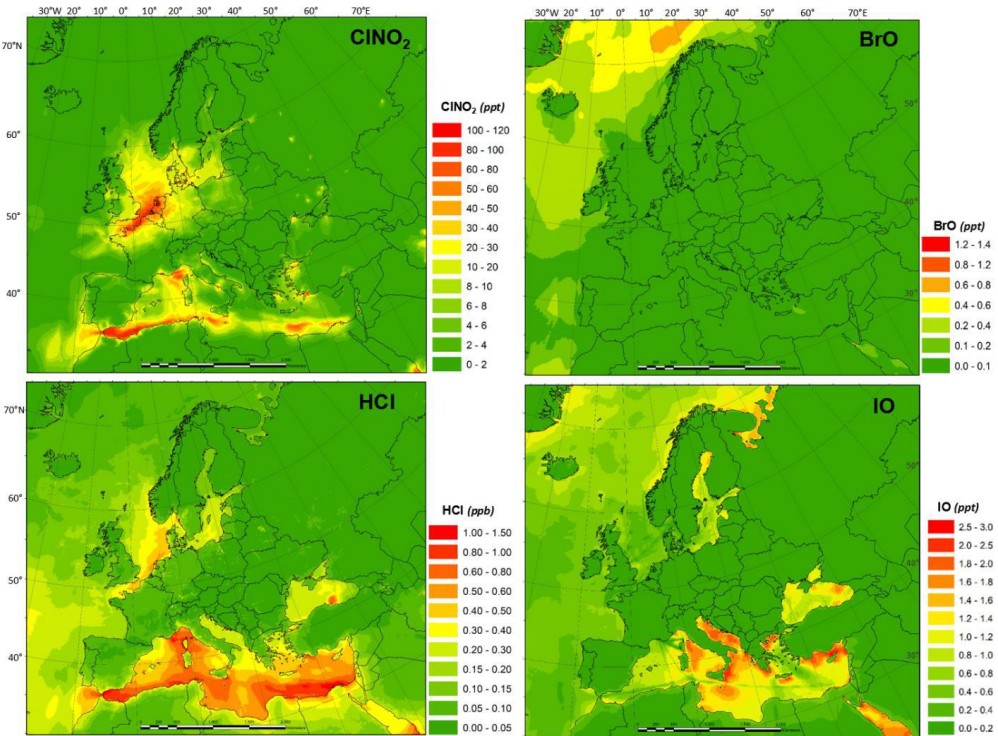

Figure 2. Monthly average $ClNO_2$, HCl, BrO, and IO concentration in the HAL simulation.

The spatial distributions of key halogen species are shown in Fig 2. The HAL simulation with

full halogen chemistry simulates generally higher $ClNO_2$ levels (with the highest average value

of 113.1 pptv) along the coast of the Mediterranean Sea and the North Sea with some influence





into continental Europe, especially in Germany. The simulated HCl shows a similar pattern to that of $ClNO_2$ but with much higher concentration (>10 times higher). The predicted BrO levels over Europe are low (average value ~0.08 pptv) with the largest predicted value occurring

within the Arctic circle. The predicted IO peaks over the Mediterranean region with a maximum value of 2.7 pptv.

Direct measurements of halogen species are very scarce and not available for the period covered in the present study (July 2016). Since a direct comparison is not possible, here we present a comparison of the simulated concentrations with observations from previous studies (Table 3),

to provide an approximate assessment of the representation of halogen species in the HAL simulation of the CMAQ model over Europe.

Table 3. The comparison of observed and simulated halogen species

| Location | Species | Observation [*] | Simulation [#] |
|---|---|---|---|
| Hessen, Germany [a] | $ClNO_2$ | 800.0 | 209.7 |
| London, England [b] | $ClNO_2$ | 724.0 | 806.3 |
| Mace Head, Ireland [c] | BrO | 6.5 | 1.8 |
| Brittany, France [d] | BrO | 7.5 | 0.2 |
| Dead Sea [e] | BrO | 100.0 | < 0.1 |
| Mace Head, Ireland [f] | IO | 4.0~50.0 | 3.1 |
| Brittany, France [g] | IO | 7.7~30.0 | 1.7 |
| Dagebull, Germany [h] | IO | 2.0 | 4.1 |

[*]: Maximum value (pptv).
[#]: Maximum value (pptv) from the HAL simulation.
a: Phillips et al. 2012.
b: Bannan et al., 2015.
c: Saiz-Lopez et al., 2004.
d: Mahajan et al. 2009.
e: Matveev et al., 2001; Holla et al., 2015.
f: Allan et al., 2000; Commane et al., 2011.
g: Britter et al., 2005; Furneaux et al., 2010.
h: Peters et al., 2005.

Numerous $ClNO_2$ measurements have been reported around the globe which show that $ClNO_2$ is ubiquitous in the boundary layer with maximum values ranging from hundreds to thousands

pptv in polluted coastal (Osthoff et al., 2008; Wang et al., 2016) and continental regions (Tham et al., 2016; Thornton et al., 2010). Two campaigns have been conducted in Europe. Phillips et al. (2012) reported a maximum of 800 pptv $ClNO_2$ in Hessen, Germany where CMAQ predicts a





concentration of 209.7 pptv. Bannan et al. (2015) observed a peak value of 724 pptv in London where CMAQ predicts a concentration of 806.3 pptv. Simulations with the GEOS-Chem model

(Sherwen et al., 2017) reported maximum values of 110 pptv and 140 pptv at Hessen and London, respectively. CMAQ predicts higher values of $ClNO_2$ and are closer to the observations compared to the GEOS-Chem model, probably due to the finer grid resolution and different uptake coefficient for heterogeneous hydrolysis of $N_2O_5$ in CMAQ which facilitates the model in capturing the local formation of $ClNO_2$.

BrO measurements have been reported at ground-based sites and during the ship cruises which generally demonstrate a range of 0.5 to 2.0 pptv maximum values for land measurements and 3.0 to 3.6 pptv for ship measurements (Saiz-Lopez and von Glasow, 2012). BrO observations have been reported at several coastal sites in Europe. BrO level of up to 6.5 pptv (Saiz-Lopez et al., 2004) and 7.5 pptv (Mahajan et al., 2009) were reported in Mace Head and Brittany,

respectively. CMAQ predicts 1.8 pptv and 0.2 pptv at those locations, which are lower than the measurements. Sherwen et al. (2017) also predicted similar values with a maximum of 0.8 pptv in Mace Head and 0.5 pptv in Brittany. An extremely high level of BrO, ~100 pptv, was observed over the Dead Sea (Matveev et al., 2001; Holla et al., 2015). CMAQ is not able to reproduce such a high level of BrO due to the lower bromide content in typical ocean water

(which was used in the present study for the Dead Sea) compared to the exceptionally high bromide content in Dead Sea (Tas et al., 2006; Sarwar et al., 2015).

Global measurements of IO show that the IO levels observed by ground-based campaigns were generally between 0.2 and 2.4 pptv while those by ship measurement were ~3.5 pptv (Saiz-Lopez and von Glasow, 2012). Observations of IO have also been conducted in Europe.

Maximum IO levels of 4.0~50.0 pptv were measured at Mace Head (Allan et al., 2000; Commane et al., 2011). CMAQ predicts a value of 3.1 pptv at Mace Head while GEOS-Chem predicted a value of 0.6 pptv (Sherwen et al., 2017). In Brittany, up to 7.7~30.0 pptv of IO were observed by Bitter et al. (2005) and Furneaux et al. (2010). CMAQ predicts 1.7 pptv of IO at Brittany and Sherwen et al. (2017) predicted 0.07 pptv. A maximum IO concentration of 2.0

pptv was reported in Dagebull (Peters et al., 2005), and CMAQ predicts 4.1 pptv at that site, while GEOS-Chem predicted 1.8 pptv (Sherwen et al., 2017).

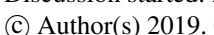



### 3.3 Influence of halogen chemistry on the atmospheric oxidation capacity

3.3.1 Monthly average

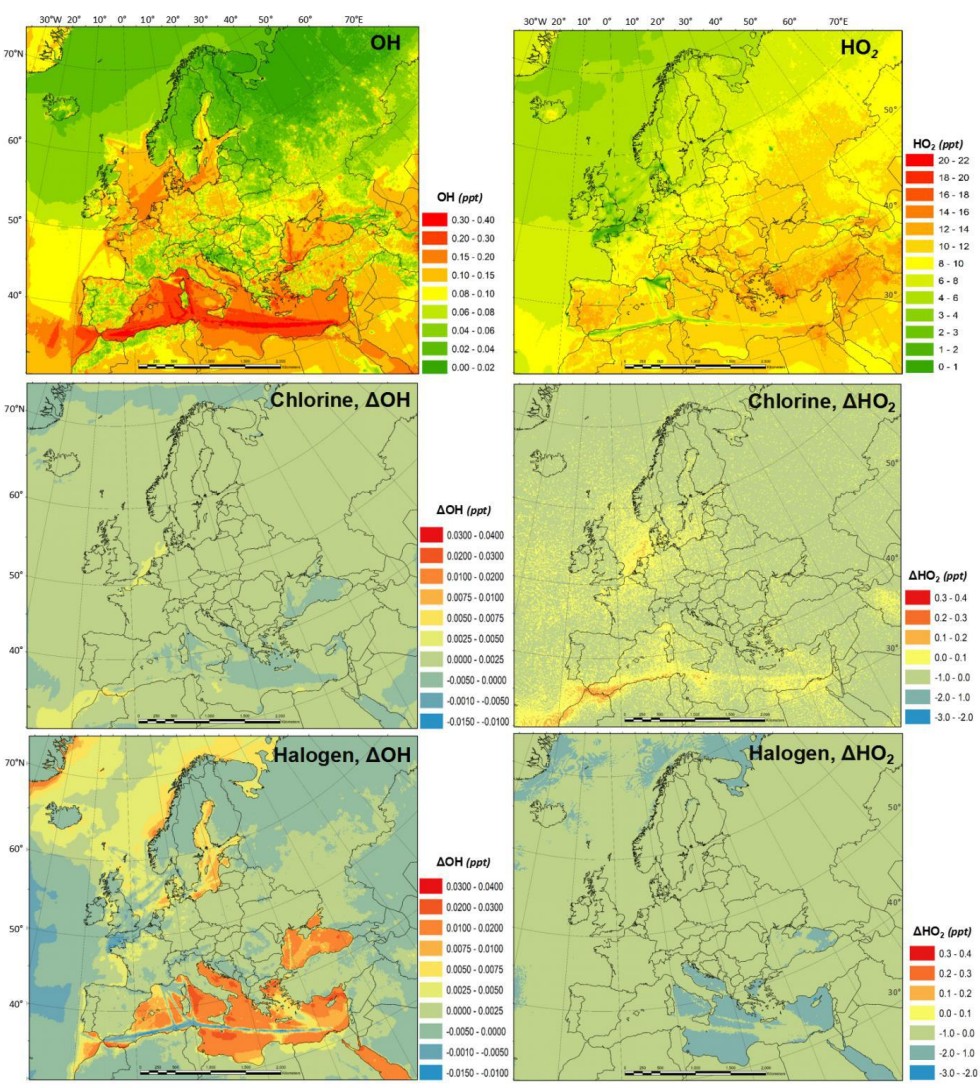

260   Figure 3. Monthly average OH and $HO_2$ concentration in the BASE simulation, and changes due to chlorine (CHL)

and full halogen chemistry (HAL).

Fig. 3 shows the monthly average concentrations of the OH and $HO_2$ radicals predicted by the
BASE simulation and the impact of chlorine chemistry (CHL-BASE), and the full halogen
chemistry (HAL-BASE), on the simulated OH and $HO_2$ levels. In the BASE simulation, the





highest OH concentration levels are predicted over the oceans especially along ship tracks, with a maximum value of 0.4 pptv. The chlorine chemistry slightly increased the OH level over most of the domain by up to 0.008 pptv. The impact of the halogen chemistry has competing effects on OH concentrations with a maximum increase of 0.03 pptv and a reduction of up to -0.01 pptv. The average concentration change was slightly positive (0.002 pptv or ~0.5%). The BASE

simulation predicts the highest values of $HO_2$ over the Mediterranean Sea with a maximum value of 20.6 pptv. The chlorine chemistry increases the $HO_2$ level in the areas of elevated $ClNO_2$ predictions (Fig. 2 and Fig. 3). The further addition of halogen chemistry lowers the $HO_2$ with a maximum effect of -3.0 pptv compared to the BASE simulation. The effect of halogen chemistry on OH and $HO_2$ is the combined effect of the following three pathways: (1)

conversion of $HO_2$ to OH via XO (R3 and R4), in which $HO_2$ decreases and OH increases; (2) reduction of of $O_3$ (R1 and Figure 7) and the reduced production of OH by $O_3$ photolysis, in which both the OH and $HO_2$ decrease; and (3) increase of $NO_2$ (R2 and Figure 7) and the enhanced consumption of OH by the reaction with $NO_2$, in which both OH and $HO_2$ decrease. Pathway (3) is particularly evident along the ship tracks in Mediterranean Sea. The significant

decrease of $HO_2$ across the domain and the marginal decrease of OH in some grid cells along with the noticeable increase in other cells suggests that the $HO_2$/OH ratio was significantly reduced by the halogen chemistry.

     Sarwar et al. (2015) reported a small overall decrease of OH (1%) and a significant decrease of $HO_2$ (11%) in the Northern Hemisphere due to the bromine and iodine chemistry. Their results

suggest a considerable reduction of the $HO_2$/OH ratio which is consistent with the present study. Muñiz-Unamunzaga et al. (2018) found a slight increase of diurnal OH (1-2%) and a noticeable decrease of $HO_2$ (4%) leading to a decrease of $HO_2$/OH in Los Angeles, California. Sherwen et al. (2017) suggested that OH was reduced across their European domain due to the halogen chemistry. They concluded that the shift of $HO_2$ to OH by XO could not compensate for the

decrease of OH due to the loss of $O_3$. The discrepancy between their study and the present one is difficult to deduce since it is probably strongly conditioned by differences in the emission inventory and/or the different spatial resolution used in their study (~25 km) and the present study (12 km).



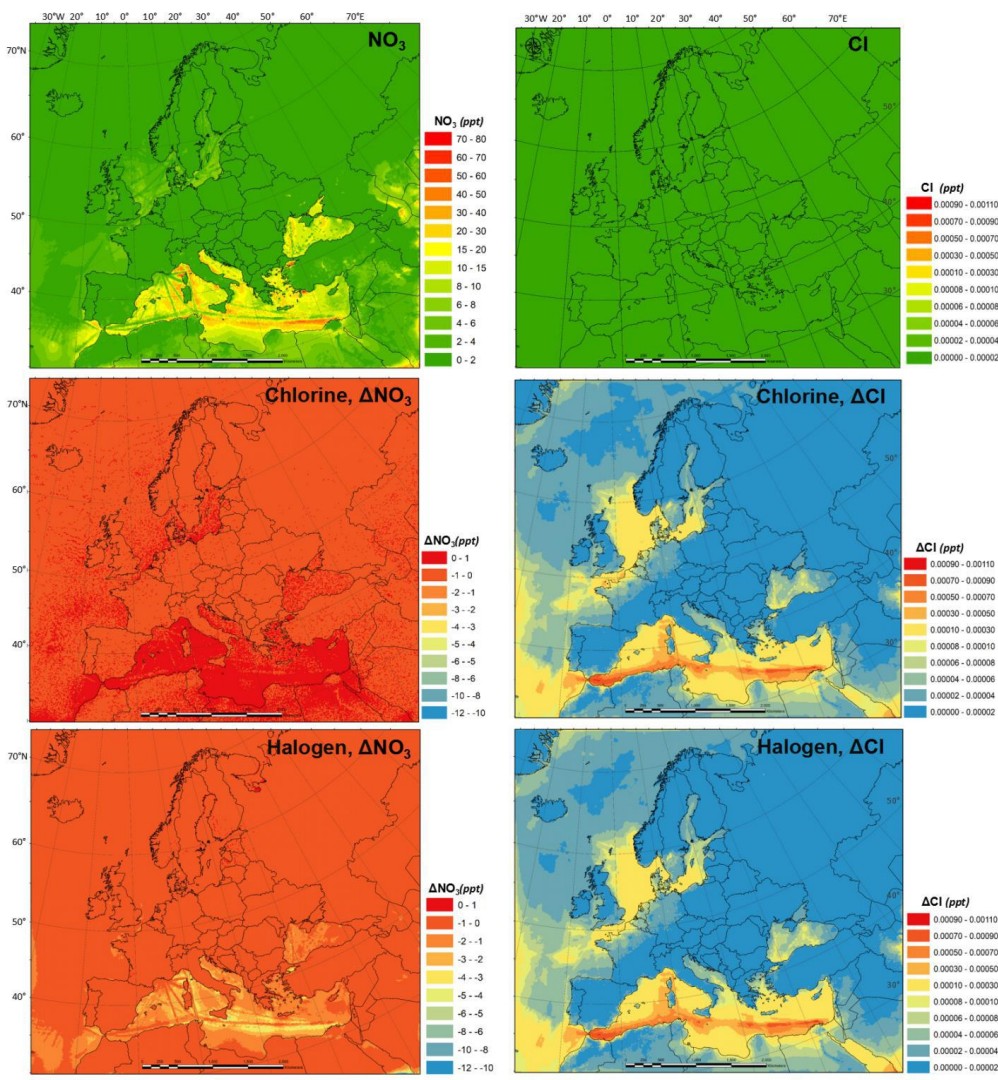

Figure 4. Monthly average NO₃ and Cl radical concentrations in the BASE simulation, and changes induced by chlorine (CHL) and full halogen chemistry (HAL).

Fig. 4 presents the monthly average prediction of NO₃ and Cl radicals in the BASE scenario and the influence of chlorine (CHL-BASE) and halogen chemistry (HAL-BASE) on the levels of NO₃ and Cl. The BASE simulation predicted relatively high NO₃ concentrations over the Mediterranean Sea along the busy shipping tracks. Although concentrations as high as 72.4 pptv are found, the majority of   oceanic regions have concentrations in the range of 10 to 40 pptv and between 0-4 pptv over land. The chlorine chemistry slightly increases the NO₃ radical due to





the increase of the $O_3$ (see section 3.4). In contrast, the halogen chemistry considerably reduces $NO_3$ concentrations by as much as -11.1 pptv at nighttime over the Mediterranean Sea. Muñiz-

Unamunzaga et al. (2018) reported a 20-50% (2-4 pptv) decrease of $NO_3$ radical in Los Angeles, California when considering the halogen chemistry. Our study, along with the previous work, highlights the vital role of halogen chemistry in the nighttime chemistry.

In the BASE simulation, the Cl concentration was negligible because there was no relevant chlorine source incorporated in the CMAQ model. The CHL simulation contains the production

of $ClNO_2$ and its subsequent photolysis which increases the Cl concentration of as high as $9.3 \times 10^{-4}$ pptv.    The HAL simulation predicted a very similar magnitude and spatial distribution of chlorine concentration. Sherwen et al. (2017) reported Cl concentrations less than $1.4 \times 10^4$ atom cm$^{-3}$ ($\sim 5.6 \times 10^{-4}$ pptv) over Europe, which is lower than but comparable to our prediction. Hossaini et al. (2016) reported over $1.0 \times 10^4$ atom cm$^{-3}$ ($\sim 4.0 \times 10^{-4}$ pptv) of chlorine over Asia,

Europe and North America, with a maximum of $8.5 \times 10^4$ atom cm$^{-3}$ ($\sim 3.4 \times 10^{-3}$ pptv), using a global chemical transport model (TOMCAT) that incorporated chlorine sources from sea salt dechlorination, coal and biomass burning, oxidation of natural and anthropogenic chlorocarbon, and heterogeneous reactions on sea salt and sulfate aerosol.

The current study and the previous works simulated a broad range of the surface Cl

concentrations although they were all within the scope of the reported observed (observation-based calculation) values of $10^3$ to $10^5$ atom cm$^{-3}$ ($\sim 4.0 \times 10^{-5}$ to $4.0 \times 10^{-3}$ pptv) according to the review of Saiz-Lopez and von Glasow (2012). In light of the considerable variation of observed and model predicted Cl level, further study may be needed to comprehensively evaluate the significant role of Cl in the troposphere.




### 3.3.2 Monthly average of daily-maximum

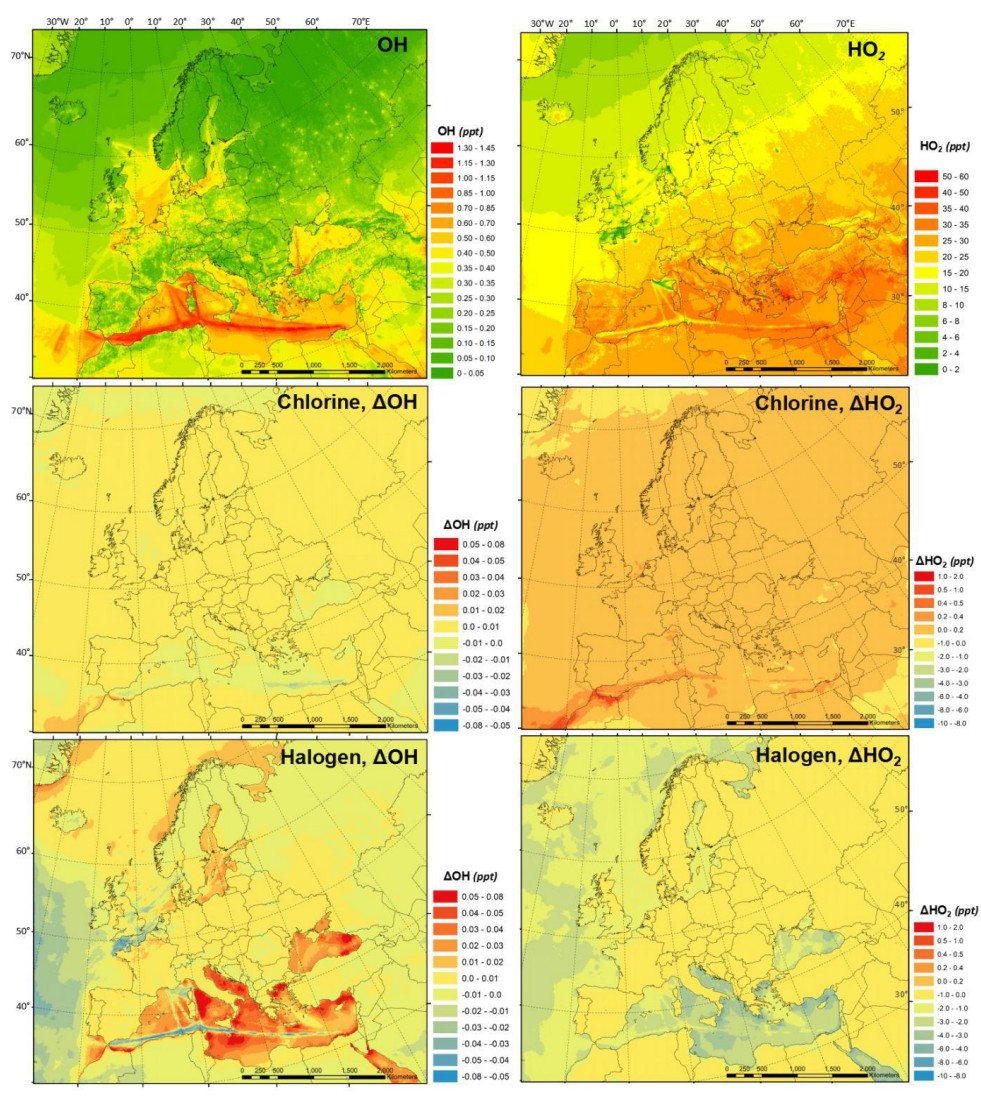


Figure 5. Monthly average of daily-maximum concentrations of OH and HO₂ in the BASE simulation, and changes

due to chlorine (CHL) and full halogen chemistry (HAL).



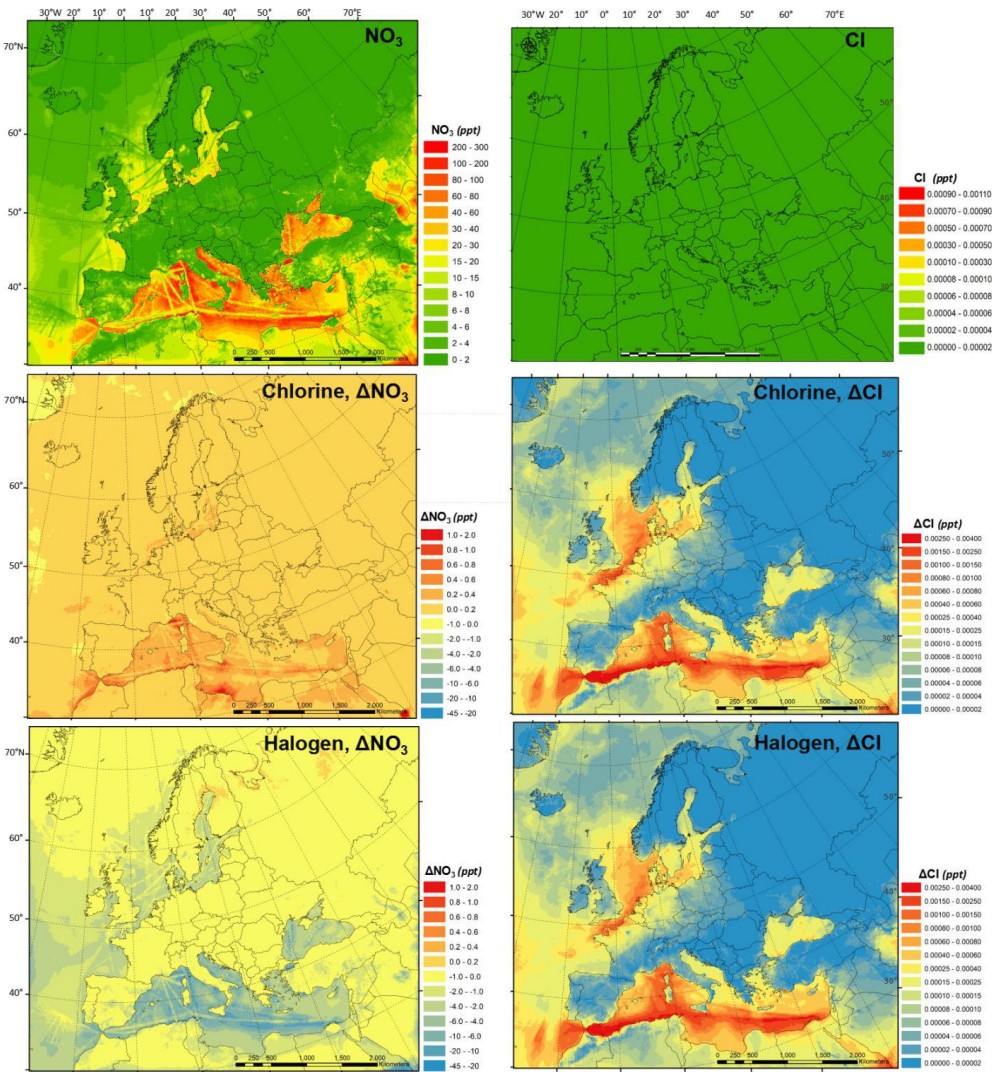

Figure 6. Monthly average of daily-maximum concentrations of NO₃ and Cl radical in the BASE simulation, and

changes induced by chlorine (CHL) and full halogen chemistry (HAL).

Fig. 5. and Fig. 6 demonstrate the monthly average of the daily-maximum concentrations of OH, HO₂, NO₃, and Cl in the BASE simulation and also the impact of chlorine (CHL-BASE) and the halogen chemistry (HAL-BASE). The maximum values of OH, HO₂, and Cl were predicted during the daytime but they peak at different hours with Cl in the early morning and OH and

HO₂ later in the day, while the highest levels of NO₃ radical were simulated during night-time.



The monthly average of daily-maximums of OH, HO$_2$, NO$_3$, and Cl (Fig. 5 and Fig. 6) have similar spatial pattern and higher concentrations (or changes of concentration) compared to those of the monthly averages (Fig. 3 and Fig. 4). The monthly average of daily-maximum OH (HO$_2$ and NO$_3$) radical is about 4 (3 and 3) times of the monthly average OH (HO$_2$ and NO$_3$) concentration in the BASE simulations, while both the monthly average of daily-maximum and monthly average Cl shows negligible values. The monthly average of daily-maximum changes of OH (HO$_2$) concentration due to the chlorine and halogen chemistry has magnitude of -0.06 to 0.06 pptv (-10.0 to 2.0 pptv), which is much wider than that of the monthly averages, i.e., -0.015 to 0.04 pptv (-3.0 to 0.4 pptv). For the NO$_3$ (Cl) radicals, the magnitude of changes in monthly average is -12.0 to 1.0 pptv (0.0 to 0.001 pptv) while that in monthly average of daily-maximums is -45.0 to 2.0 pptv (0.0 to 0.004 pptv).

The significant effects of halogen chemistry on the daily-maximums of OH, HO$_2$, NO$_3$, and Cl radicals highlight the role of halogen chemistry in regulating the atmospheric oxidation capacity throughout the day with the highest effect on Cl in the early morning, maximum effects on OH and HO$_2$ in daytime, and largest effect on NO$_3$ at night.





### 3.4 Impact of halogen chemistry on regulated gaseous air pollutants

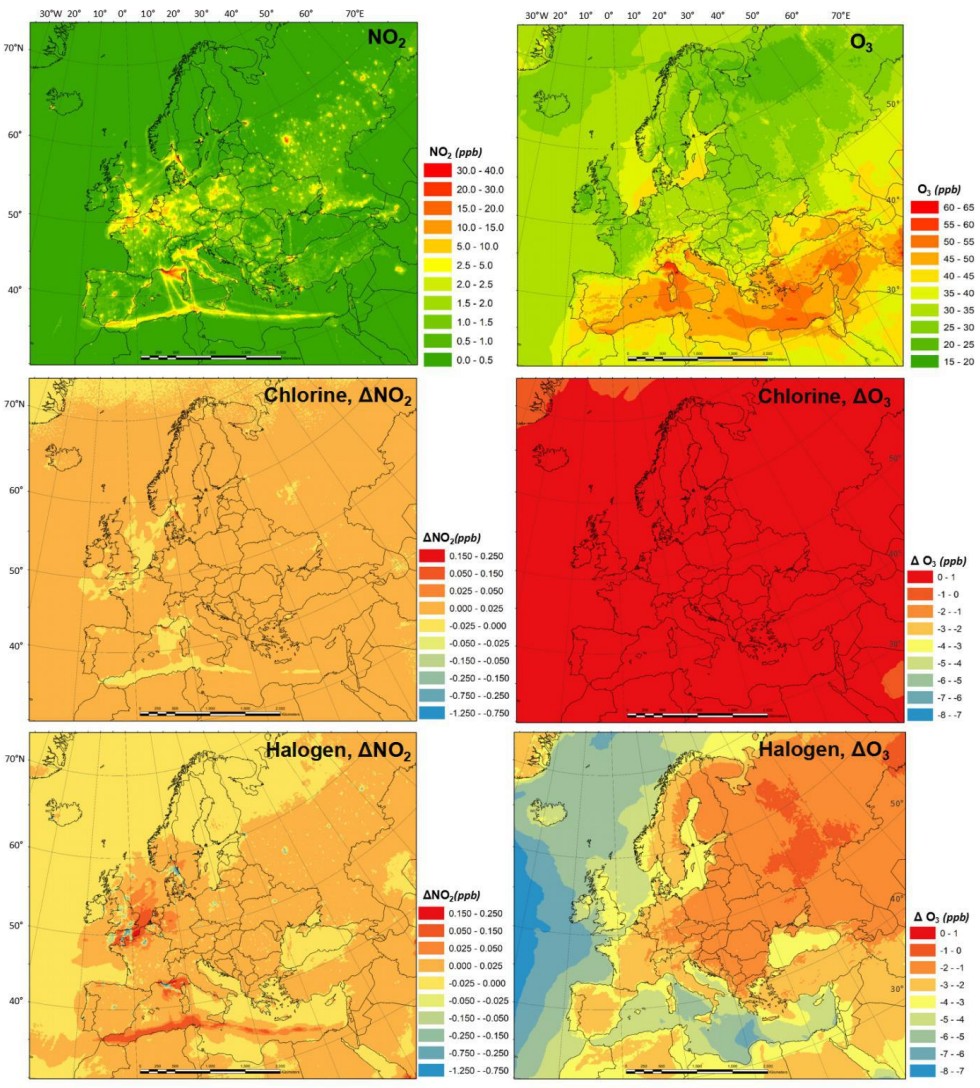

Figure 7. Monthly average $NO_2$ and $O_3$ concentration in the BASE simulation, and changes induced by chlorine (CHL) and full halogen chemistry (HAL).

The monthly average modeled $NO_2$ and $O_3$, two major gaseous air pollutants in Europe, and the effect of chlorine and halogen chemistry on the two regulated gaseous species were shown in Fig. 7. The BASE simulation produced many hot spots of $NO_2$ over Europe, in the vicinity of the major cities and the ship trajectories. The chlorine chemistry slightly increases the level of $NO_2$





(by up to 0.10 ppbv) in the majority of the domain since (1) the production and the subsequent photolysis of $ClNO_2$ recycles the $NO_x$ extending its lifetime, which increases both $NO_2$ and NO (Fig. 7 and S2) and (2) the increased $O_3$ level (Fig. 7) enhances the transformation of NO to $NO_2$, which increases $NO_2$ and decreases NO (Fig. 7 and S2). Some grid cells show a decrease of $NO_2$ (up to -0.05 ppbv) because the enhanced oxidative capacity (Section 3.3) promotes the cleansing of $NO_2$ via OH forming $HNO_3$. The full halogen chemistry enhances $NO_2$ (up to 0.22 ppbv) over the North Sea and the Mediterranean Sea and decreases $NO_2$ (as much as -1.1 ppbv) in the most polluted hot spots. The increase of $NO_2$ occurs by the reactions of XO with NO which increases $NO_2$ and decreases NO. Meanwhile, in the most polluted regions, the $NO$-$NO_2$ balance is predominantly controlled by the reactions of NO with $HO_2$ and $O_3$. With the decrease of $HO_2$ and $O_3$ due to the halogen chemistry, the transformation of NO to $NO_2$ is reduced which leads to decreasing $NO_2$ and increasing NO.

The monthly average $O_3$ concentration over Europe from the BASE simulation was relatively high (up to 60.9 ppbv) especially over the southern Europe, where higher temperature and more intensive radiation promote the formation of this secondary pollutant. Chlorine chemistry increases $O_3$ levels with a maximum increment of ~0.9 ppbv. The full halogen chemistry, instead, decreases $O_3$ throughout the domain with a maximum reduction of -7.9 ppbv. On average, the halogen chemistry reduces $O_3$ concentration by more than 3.0 ppbv in coastal Europe and by over 2.0 ppbv over western and central Europe (nearly one thousand kilometers from the ocean). Our model simulation highlights the fact that halogen chemistry has a large impact on $O_3$ concentrations over the oceanic areas and a moderate impact on $O_3$ over coastal and continental regions of Europe.

Muñiz-Unamunzaga et al. (2018) reported a decrease of 2.0 ppbv $O_3$ in the inland areas of the western US (several hundreds of kilometers from the ocean) and a reduction of 2.5-5.0 ppbv $O_3$ in the coastal regions due to the full-halogen chemistry. Sarwar et al. (2015) suggested that the inclusion of halogen processes, which is the same as that in the present study, reduced $O_3$ concentrations by 2.0-4.0 ppbv over most of the terrestrial regions in the North Hemisphere, and over 6.0 ppbv in some coastal areas. Sherwen et al. (2017) used a revised version of GEOS-Chem (Sherwen et al., 2016) with halogen chemistry to show substantial reductions in $O_3$ over



Europe with an average reduction of -13.5 ppbv in the domain and a maximum of -28.9 ppbv in some locations.

**3.5 Implications for policy assessment**

The current air quality management in Europe has two main objectives: (1) to protect human health and (2) to protect the environment. While many plans and measures have prioritized PM or $NO_2$, policies to reduce $O_3$ concentrations are still needed (EEA, 2018a). The WHO Air Quality Guidelines value for $O_3$ (maximum daily 8-hour mean of 100 $\mu g \cdot m^{-3}$) was exceeded in 96% of all the reporting stations in Europe, although this is especially true for the areas near the Mediterranean Sea. According to the EEA latest report, 12% of the EU-28 urban population is exposed to $O_3$ concentrations above the EU target value threshold (maximum daily 8-hour mean of 120 $\mu g \cdot m^{-3}$ not to be exceeded on more than 25 days/year, as set out by the Directive 2008/50/EC) in 2016. Apart from significant potential health effects (Jerrett et al., 2009; Malley et al., 2017), $O_3$ is also known to have a negative impact on vegetation (Mills et al., 2011). The target value for the protection of vegetation (18,000 $\mu g \cdot m^{-3} \cdot h$ accumulated over May to July), based on the Accumulated Ozone exposure over a Threshold of 40 ppbv index (AOT40 index, Fig. 8), was exceeded in about 31 % of all agricultural land in all European countries. The critical level for this pollutant (10,000 $\mu g \cdot m^{-3} \cdot h$ accumulated over April to September) was exceeded in 60% of the total forest area of the continent in 2016 (EEA, 2018a).



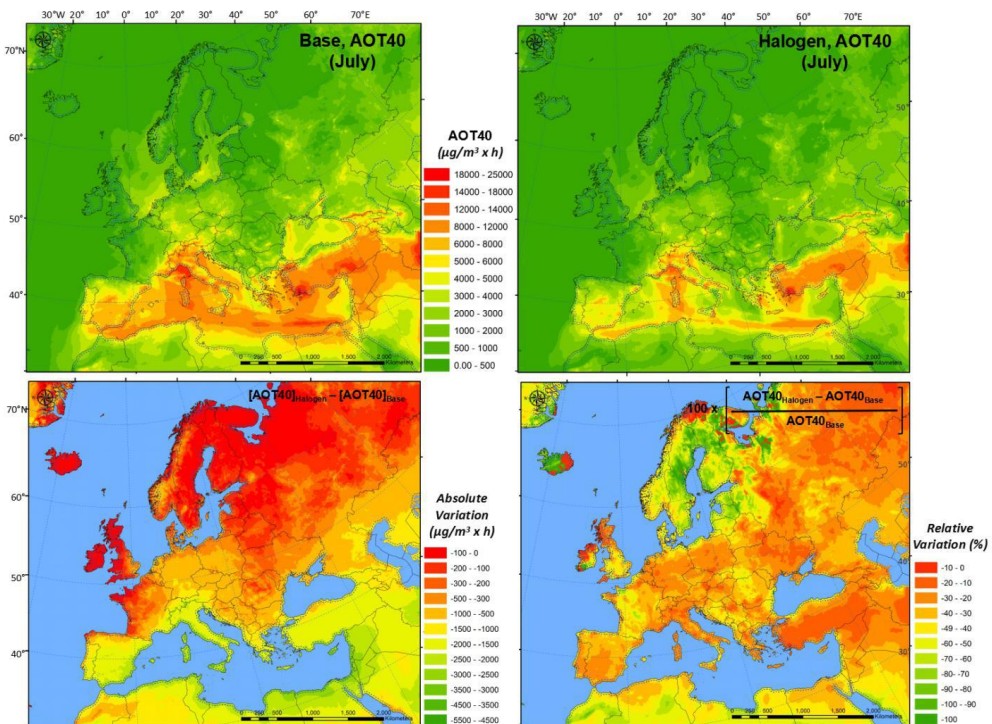

Figure 8. Monthly average AOT40 index in the BASE and HAL simulations, and absolute and relative changes between the two simulations.

We find that halogen chemistry strongly affects ambient $O_3$ concentration and may need to be considered in the formulation of plans and strategies for $O_3$ non-attainment areas. We see differences between BASE and HAL simulations (over land in July 2016) as high as 12% and 36% for the number of days with daily maximum 8 h $O_3$ over 120 $\mu g \cdot m^{-3}$ and the AOT40 respectively (Fig. S3 and Fig. S4). Furthermore, we notice strong regional differences, mainly between coastal and inland areas. The considerable effect of halogen chemistry on air quality implies the need to improve the robustness and accuracy of modeling tools to design customized policies to control $O_3$.

In Section 3.3 and 3.4, we have also discussed the effect of halogen chemistry on the partitioning of $OH/HO_2$ and $NO/NO_2$. The budgets of $HO_x$ and $NO_x$ are key parameters to accurately simulate the formation of $O_3$ and its response to the reductions of the precursors, namely $NO_x$ and VOCs (e.g., Li et al., 2018). Air quality models are predominantly used to formulate air pollution control policy by examining the responses of $O_3$ levels to various





reduction rates of NO$_x$ and/or VOCs. These models do not include the comprehensive halogen chemistry, potentially leading to unrealistic simulation of O$_3$ concentration responsiveness to the predicted NO$_x$ and/or VOCs emission changes in Europe.

This study also demonstrates that chlorine chemistry enhances the formation of O$_3$. The current policy is only designed to control the long-lived chlorinated species (Hossaini et al., 2015), but

not the reactive chlorine species, e.g., HCl, chloride, and short-lived chlorocarbons, from the coal burning, biomass burning, and industrial activities. The coal-fired power plants in EU (EEA, 2018b; Kuklinska et al., 2015) can potentially provide chlorine sources, making the implications of halogen chemistry even more relevant.

**4. Conclusion**

We applied the CMAQ model with comprehensive halogen chemistry (Cl, Br and I) to conduct high-resolution simulations for examining the impact of halogen chemistry on air quality over Europe.

The comparison of model results with observations from 465 monitoring sites indicates that the

CMAQ model is capable of reproducing the concentrations and temporal variations of air pollutants over Europe and can be employed to study the impact of halogen chemistry in Europe. The comparison of predicted halogen species concentrations with measurements suggests that CMAQ model is able to predict observed levels of chlorine and iodine species although it underestimates bromine species.

The chlorine chemistry enhances the atmospheric oxidation capacity by significantly increasing the level of Cl radical and increases the levels of OH, HO$_2$, NO$_3$, O$_3$, and NO$_2$. The combined halogen chemistry marginally increases the level of OH and reduces HO$_2$, NO$_3$, and O$_3$. The impact of halogen chemistry on ambient concentration of NO$_2$ is smaller but non-negligible.

Halogen chemistry significantly influences the atmospheric oxidation capacity throughout the

day by imposing the highest effect on Cl in the early morning, maximum effects on OH and HO$_2$

in daytime, and largest effect on NO$_3$ at night. Halogen chemistry can have a strong influence on atmospheric composition over oceanic and coastal regions but also some noticeable impacts over continental Europe. This study highlights the potential benefit of incorporating halogen chemistry into air quality models for policy development.

Although the incorporation of the halogen chemistry may improve the capabilities of 3D Eulerian chemical transport models, we acknowledge that large uncertainties still exist in the assessment of halogen chemistry impact due to emission inventories, model configuration (e.g., grid size), chemical mechanism, etc. Further field, laboratory, and theoretical studies are needed to constraint modeling studies for evaluating the impacts of halogen chemistry on air quality and
for assessing air quality policy implications.

*Data availability.* The data used and demonstrated in this study are available upon the request to the corresponding author.

*Author contributions.* AS-L designed research. GS and BG conducted the CMAQ modeling.
RB, DP, JD, GS, BG, QL, and AS-L analyzed the results. QL, RB, GS, and AS-L wrote the paper with contributions from all authors.

*Competing interests.* The authors declare that they have no conflict of interest.

*Acknowledgment.* This study has received funding from the European Research Council Executive Agency under the European Union´s Horizon 2020 Research and Innovation
programme (Project 'ERC-2016-COG 726349 CLIMAHAL').

### DISCLAIMER

The views expressed in this paper are those of the authors and do not necessarily represent the views or policies of the U.S. EPA.




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
