# Peer review of "Impact of halogen chemistry on summertime air quality in coastal and continental Europe: application of CMAQ model and implication for regulation"

_Atmospheric Chemistry and Physics, 2019_

## Referee Comment (RC1) · Anonymous Referee #1 · 20 May 2019

General comments

This work presents the evaluation of the halogen (Cl, Br, I) chemistry scheme from Sarwar et al [2015] within the CMAQ model on a regional scale (12km) for the month of July over Europe. The authors present an evaluation of halogen chemistry's impacts on air-quality metrics by comparing two runs, with chlorine ("CHL") or all halogens (HAL), against a model run without halogen chemistry ("BASE"). The focus is on how halogen chemistry impacts ozone in the European summer.

The authors argue the novel aspect of this study derives from the high resolution (12km) and focus on air quality impacts over Europe. However, this novelty is somewhat challenged by the extensive referencing of a study in another model that considered also considered halogen chemistry within a nested model over Europe at ∼25km during the summer too [Sherwen et al 2017]. Furthermore, this work seems to omit reference to the follow-on work using the same nested model ∼25km for other seasons [Sommariva et al 2018] or using the model in a global for an entire year at 12km [Hu et al 2018]. It is not clear what the authors are suggesting is the main differences in between the nested global vs. nested hemispheric approaches, apart from the resolution.

Furthermore when considering the period of study: Halogen chemistry can be very seasonally dependent in Europe (e.g. ClNO2 - See Sommariva et al [2018]). So considering the existing literature on modelling halogens, further model runs should at least be presented for a winter month to give an equivalent novel value on understanding the air-quality impacts.

Some opportunities to make this work more novel seem to have been missed. For instance, coarse two bin comparisons ("coastal" vs. "inland") are made with regional ozone and NO2 observations instead of comparisons that show if the model captures chemical/physical processing (e.g. diel plots by the hour). Oxidants are analysed as "snapshots" of peak or average concentrations, rather than considered in terms of a given hour. Considering the diel cycle of oxidants by hour over Europe (e.g. contribution of Cl in the morning vs. OH at midday, then NO3) and how halogens effect this would really elevate this manuscript.

The description of the model setup needs to be clearer. From reading the manuscript, I think I correctly understand that the configuration uses a regional grid for Europe and then sets "boundary conditions" outside this grid from two additional hemispheric runs. Please add more information to the model configuration section and clarify this.

Writing is often verbose and would often benefit from re-wording to make the manuscript more concise. The introduction especially would benefit for re-writing for

flow and clarity of the various different model studies cited. References (e.g. for observations) could be more up to date.

Specific comments

Page 1 - Lines 15-35 (Abstract) Please add headline numbers for changes seen in this manuscript to abstract.

What is the net effect of the domain on OH/HO2? The effect of halogens on OH /HO2 in box models and global models has been discussed at length by Stone et al. [2018].

Page 2 - Line 35 - 41 Earlier references should be used here to give fair credit to the original work on this, instead of recent reviews.

It would be cleaner and more instructive to the reader to include the following sentence, rather than just citing a recent review.

"The chemistry of halogens in the troposphere has been described in detail in recent reviews (Saiz-Lopez and von Glasow, 2012; Simpson et al., 2015), so we just briefly outline it here."

Additionally, the effect of halogen nitrate hydrolysis on decreasing ozone production should be mentioned here as it has been shown to have a larger impact on ozone than increased loss [Schmidt et al 2016].

Page 2 - Line 46 "methane ch4" should read "methane (CH4)"

Page 2 - Line 52-55 "Evaluation of the complex role of halogen chemistry in air quality requires the employment of advanced, high-resolution chemical transport models"

Has this argument really been evidenced here? Could not the processed discussed be captured by existing coarse resolution approaches? Are the uncertainties on halogen modelling substantially small enough that horizontal resolution of models is the main limitation for the fidelity of simulation to observations and capturing chemical and physical processes?

Page 2 Line 56-61 Should not the faraday discussions paper you mentioned earlier be included here as you say in the manuscript it covered ClNO2 too? Other model studies have looked at this too and should be included here also. Adding "e.g. X et al., Y et al." to illustrate the reader that a couple of examples have been given would also be appropriate. There are other points in the manuscript where this would be appropriate too.

Page 3 - 61-71 It is hard for the reader to follow the way in which the previous work is being discussed. The authors have referred to another nested regional study in GEOS-Chem in the final intro paragraph [Sherwen et al 2017], but then did not include it in the discussion of existing regional modelling work here.

Page 3 - Line 71 It is arguable that 4km is a higher resolution. However, is a 4km horizontal resolution accepted to be sufficient to represent the processes going on in a city? I would suggest updating as follows:

From

"halogen sources on air quality at a city scale (4 km resolution) in Los Angeles, California, US."

To

"halogen sources on air quality at a resolution of 4 km in the city of Los Angeles (California, US). "

Page 3 - Line 74 A URL in brackets is not an appropriate format here. If a URL must be used please include a reference to the EEA and a data accessed for the data. Referencing an EEA report would be preferable.

Page 3 - Line 72 to 82 It is hard for the reader to follow the semantic. The previous study is also a domain based study over Europe using boundary conditions from a model with a larger (global) domain with offline meteorology. Are the authors arguing that the main difference between the 12x12km CMAQ approach presented here and the existing

work, at the coarser resolution (25x25km), is just resolution? Both models use similar halogen chemistry are nested within larger domains, correct? Why is so much change in simulation skill expected to be seen between 12x12 km and 25x25km? Or would a higher resolution, say 4x4 km [MunÌČiz-Unamunzaga et al., 2018], be required to notable gains in capturing processes or differences? This leads to a more philosophical question: is it a model resolution or processes holding back science currently? Would not other uncertainties in halogen chemistry be greater than the difference caused by a change in resolution too? (e.g. emissions developed for coarser resolutions or new developments in the representation of halogen chemistry in models - Xuan et al [2019])

Page 3 - Line 84 Is "instrumented" the appropriate word here? "Including" would be a better choice or "which includes".

Page 3 - Line 85 Why is "state-of-the-art" used here? It seems unnecessarily verbose, especially when referring to a paper that is at least four years old in a fast-moving part of the literature.

Page 4 - Line 90-96 The meteorology is offline? Or is CMAQ being run in coupled mode? Please explicitly state if the meteorology is offline.

Page 4 - Line 96 Why is the word "besides" used in this sentence. Please rephrase for clarity.

Page 4 - Line 111 How has coastal been defined here? More broadly, is 12km sufficient resolution to capture "coastal" effects?

Page 6 - Line 131 -132 Is this 12x12km domain nested within another domain not computed online? Or is the model run hemispherically at 12x12km here for all simulations? Please make this clearer.

Page 6 - line 143 The name "CHL" may make many readers from the Earth sciences community think of chlorophyll. I would suggest using "Cl" instead to make the paper more accessible to a broader readership.

Page 6 - Line 139 -145 Only 7 days spin up was used for a hemispheric simulation? Or is this for the European grid? Is this sufficient to ensure initial conditions are "washed" through the grid? Is there a reference showing this is sufficient? What spin up was used for "the hemispheric CMAQ simulations"/hemispheric grid? What initial conditions were used? This needs to be clearer. Global studies on halogens in CMAQ, GEOS-chem and CAM-Chem have highlighted the importance of changes in background concentrations. The relative contribution of boundary and local effects seems to be a core focus of the Sherwen et al [2017] manuscript, but not really given much attention or discussion here.

Page 6 - Line 152-160 The model is being run on a hemispheric grid of 12x12, but just analysed for the domain in Figure 1? The "Boundary conditions" are on the same grid, but global? and these are provided to the outside of the modelling domain? This needs to be explained more clearly.

Page 6 - Line 152 "Boundary conditions for the model were derived from the hemispheric CMAQ simulations."

Which hemispheric CMAQ simulations? Where have these been described?

Page 7 - Line 161-163 Please include a table that shows which boundary conditions and chemistry used in each domain for each model experiment.

"the difference between CHL and BASE simulations represents the impact of the chlorine chemistry on air quality"

Don't both of these simulations have the same boundary conditions ("Carbon Bond chemical mechanism and the chlorine chemistry")? Surely then the most this difference can show is the contribution of chlorine chemistry locally. A preferable approach would be to use a "BASE" set of boundary conditions without halogen chemistry for the "BASE" simulation.

"the difference between HAL and BASE simulations represents the effect of halogen

chemistry on air quality."

Again: according to the text, "BASE" includes the effects of chlorine globally as it includes chlorine in boundary conditions ("Carbon Bond chemical mechanism and the chlorine chemistry"). Therefore "BASE" - "HAL" is giving the effects of halogens minus the global effect of chlorine.

Although locally chlorine can provide an oxidant effect and lead to ozone formation, it also can act as a sink for ozone through the loss of chlorine nitrates on a global scale. This has been discussed in some of the global modelling papers cited here. What are the global effects of chlorine in this model? Can global effects be excluded here?

Page 7 - line 169 Please give a justification for the use for "within 24 km from the coast" definition. Coastal processes (notably halogens) can be confined to a very small area (e.g. macroalgae in tidal zones).

Page 7 - Line 169-171 The level of the evaluation presented needs to be increased to be in line with the high-resolution output the authors argue is notable here.

Please plot ozone and NO2 as a diel (24-hour) comparison compared with the model, preferably showing a few regions in Europe. Regions like the coastal Mediterranean should not just be lumped together with the coastal Scandinavian, as they have very different seasonal characteristics. Simply using a table to do a very coarse two bin comparison does not provide much insight and it is not really in line with the current level analysis presented in the literature (e.g. Schnell et al. 2015), instead, it smooths out the extra information gained.

Page 7 - Line 179-181 Please provide references for the proposed explanations for model bias or expand this discussion.

Page 7 - Line 184-186 Only a single table of comparisons has been provided to back up this statement. More evidence is needed. Please provide diel plots of core species (e.g. CO, NO2, O3).

[Figure]

Page 7 - Line 182 How does this comparison look on a diel basis? Is there an offset or difference in the diel cycle at certain times of the day? Does the model capture the diel cycle before or after or is there a structural issue in the model (e.g. caused by emissions or boundary layer mixing?)?

Page 11 - Line 223-255 The coastal influence or inland nature of observations should be made clearer. At the resolution presented here (12x12km) it would not be fair to expect the model to reproduce many of these observations (e.g those influenced by local emissions from the tidal zone).

Page 11 - Table 1 This table of observations does not seem in line with current literature (e.g. observations for ClNO2 are available across seasons in Northern Europe [Sommeriva et al 2018]). Please expand these comparisons.

Page 11 - Line 249 The largest IO dataset has been published since Saiz-Lopez and von Glasow [2012] by Prados-Roman [2015]. Did this not this start and end in the Mediterranean? How does the model compare against this? What other more recent datasets are there?

Page 12 - Line 261-264 Please see earlier comments about the inclusion of chlorine in boundary conditions for both "BASE" and "CHL" simulations.

Page 13 - Line 267 Please give OH units in the more commonly used units of molecules cm-3 or at least show this in brackets throughout the text. Please do the same for Cl (atoms cm-3).

Page 13 - Line 290 What about the resolution difference could cause this? What other differences could explain this? Are there any differences in the chemistry between the Sarwar et al (2015) and Sherwen et al (2017) Mechanism?

Page 15 - Line 305 "Our study, along with the previous work, highlights the vital role of halogen chemistry in the nighttime chemistry."

Which previous work? How is this chemistry constrained by lab work/observations?

[Figure]

Is it based on theoretical calculations? If So, then the uncertainty on this should be highlighted here.

page 15 Line 307 - 309 Cly is in the boundary conditions for both "BASE" and "CHL", correct? How much Cly is transported into the domain?

Page 15 - 312-318 How do these values compare against known constraints on tropospheric chlorine (e.g. Gromov et al [2018])? Are there any explanations for the differences? (e.g. It is worth noting too that Hossani et al [2016] used unrealistic anthropogenic chloride emissions - see Xuan et al [2019] for details on this)

page 15 Line 320 "The current study and the previous works simulated a broad range of the surface Cl concentrations although they were all within the scope of the reported observed (observation- based calculation) values of 103 to 105 atom cm-3 ($\sim 4.0 \times 10$-5 to $4.0 \times 10$-3 pptv) according to the review of Saiz-Lopez and von Glasow (2012)."

Why is a review that is > seven years old being used as the basis for comparison? There have been a large amount of Cly measurements since then (e.g. Gromov et al [2018], Haskins et al [2018] etc ...) and more work to constrain tropospheric Cl [Gromov et al 2018].

page 22 Fig 8 AOT40 is calculated over a growing season. Why is this shown for a single month in Fig. 8 and discussed in the text? It would be more appropriate to give output in units of exceedances for a given month as this usage of "AOT40" could mislead readers. What over relevant ozone thresholds are there? What about the particulate matter if the oxidants have changed (e.g. PM10, PM2.5)?

page 23 Lines 430-432 "These models do not include the comprehensive halogen chemistry, potentially leading to [an] unrealistic simulation of O3 concentration responsiveness to the predicted NOx and/or VOCs emission changes in Europe."

As only a coarse comparison is provided for ozone (table 1) and no diel cycles are shown it is hard to see if halogens are aiding the capture of processes seen within the

observations. This weakens the argument that halogen processes are needed to gain a "realistic" simulation in air quality models. Please back up this claim with figures.

page 24 Lines 460-465 The final paragraph of the conclusion comes across as vague. Please provide a few examples of uncertainties that are more specific than just "chemical mechanism".

References

Gromov, S., Brenninkmeijer, C. A. M., and JoÌĹckel, P.: A very limited role of tropospheric chlorine as a sink of the green-house gas methane, Atmos. Chem. Phys., 18, 9831–9843, https://doi.org/10.5194/acp-18-9831-2018, 2018.

Haskins, J. D., JaegleÌĄ, L., Shah, V., Lee, B. H., Lopez-Hilfiker, F. D., Campuzano-Jost, P., Schroder, J. C., Day, D. A., Guo, H., Sullivan, A. P., Weber, R., Dibb, J., Campos, T., Jimenez, J. L., Brown, S. S., and Thornton, J. A.: Wintertime Gas- Particle Partitioning and Speciation of Inorganic Chlorine in the Lower Troposphere Over the Northeast United States and Coastal Ocean, J. Geophys. Res.-Atmos., 123, 12897–12916, https://doi.org/10.1029/2018JD028786, 2018.

Hu, L., Keller, C. A., Long, M. S., Sherwen, T., Auer, B., Da Silva, A., Nielsen, J. E., Pawson, S., Thompson, M. A., Trayanov, A. L., Travis, K. R., Grange, S. K., Evans, M. J., and Jacob, D. J.: Global simulation of tropospheric chemistry at 12.5 km resolution: performance and evaluation of the GEOS-Chem chemical module (v10-1) within the NASA GEOS Earth System Model (GEOS-5 ESM), Geoscientific Model Development Discussions, pp. 1–32, https://doi.org/10.5194/gmd-2018-111, 2018.

Prados-Roman, C., Cuevas, C. A., Hay, T., Fernandez, R. P., Mahajan, A. S., Royer, S.-J., Galí, M., Simó, R., Dachs, J., Großmann, K., Kinnison, D. E., Lamarque, J.-F., and Saiz-Lopez, A.: Iodine oxide in the global marine boundary layer, Atmos. Chem. Phys., 15, 583-593, https://doi.org/10.5194/acp-15-583-2015, 2015.

Sommariva, R., Hollis, L. D. J., Sherwen, T., Baker, A. R., Ball, S. M., Bandy, B. J., Bell,

[Figure]

T. G., Chowdhury, M. N., Cordell, R. L., Evans, M. J., Lee, J. D., Reed, C., Reeves, C. E., Roberts, J. M., Yang, M., and Monks, P. S.: Seasonal and geographical variabil- ity of nitryl chloride and its precursors in Northern Europe, At- mos. Sci. Lett., 19, e844, https://doi.org/10.1002/asl.844, 2018.

Stone, D., Sherwen, T., Evans, M. J., Vaughan, S., Ingham, T., Whalley, L. K., Edwards, P. M., Read, K. A., Lee, J. D., Moller, S. J., Carpenter, L. J., Lewis, A. C., and Heard, D. E.: Impacts of bromine and iodine chemistry on tropospheric OH and HO2: comparing observations with box and global model perspectives, Atmos. Chem. Phys., 18, 3541-3561, https://doi.org/10.5194/acp-18-3541-2018, 2018.

Schnell, J. L., Prather, M. J., Josse, B., Naik, V., Horowitz, L. W., Cameron-Smith, P., Bergmann, D., Zeng, G., Plummer, D. A., Sudo, K., Nagashima, T., Shindell, D. T., Falu- vegi, G., and Strode, S. A.: Use of North American and European air quality networks to evaluate global chemistry–climate modeling of surface ozone, Atmos. Chem. Phys., 15, 10581-10596, https://doi.org/10.5194/acp-15-10581-2015, 2015.

Schmidt, J. A., Jacob, D. J., Horowitz, H. M., Hu, L., Sherwen, T., Evans, M. J., Liang, Q., Suleiman, R. M., Oram, D. E., Le Breton, M., Percival, C. J., Wang, S., Dix, B., and Volkamer, R.: Modeling the observed tropospheric BrO background: Im- portance of multiphase chemistry and implications for ozone, OH, and mercury, J. Geophys. Res.-Atmos., 121, 11819–11835, https://doi.org/10.1002/2015jd024229, 2016.

Wang, X., Jacob, D. J., Eastham, S. D., Sulprizio, M. P., Zhu, L., Chen, Q., Alexander, B., Sherwen, T., Evans, M. J., Lee, B. H., Haskins, J. D., Lopez-Hilfiker, F. D., Thornton, J. A., Huey, G. L., and Liao, H.: The role of chlorine in global tropospheric chem- istry, Atmos. Chem. Phys., 19, 3981-4003, https://doi.org/10.5194/acp-19-3981-2019, 2019.

---

## Referee Comment (RC2) · Anonymous Referee #3 · 21 May 2019

This paper presents a model study of the effects of halogen chemistry on the air quality of Europe. This study provides an interesting overview of the impact of halogens on ozone and other pollutants, a research question that is still open. The paper is well written and presented and I have only some minor comments (see below). Overall I think it is suitable for publication in ACP.

In Section 3.2 the CMAQ results are compared to the observations and to the GEOS-Chem results from Sherwen et al. (2017). First of all, there are other observations of ClNO2 in Europe besides those in Table 3. In fact some of these are mentioned in the Sherwen paper itself (as well as in Sommariva et al., 2018) and in Bannan et al.

[Figure]

(2017). These measurements should be included in the discussion. Second, using the maximum observed concentration is not a good metric to assess the agreement with the model. For example, the observations in Phillips et al. (2012) show quite a range of peak nocturnal concentrations of ClNO2. I would also argue that GEOS-Chem shows better agreement with the measurements than CMAQ, especially wrt ClNO2 (lines 231-233). The discussion of the model-measurements comparison is better when dealing with iodine and bromine species, but please revise Section 3.2 to be more accurate.

Figure 2 is interesting in the sense that it shows some different results from the corresponding figure 5 in the Sherwen paper especially when it comes to BrO. It looks like CMAQ is calculating lower concentrations than GEOS-Chem both for Cl and for HCl, which deserves some comment. It would also be good to include some of the European observations of HCl in this discussion. I realize that a comparison between CMAQ and GEOS-Chem is beyond the scope of this paper, but the differences in the geographical distributions of some species (and related impacts on O3 and other species) are sometimes striking and require at least a brief comment.

line 688: correct typo in name.

---

## Author Comment (AC1) · 13 Sep 2019

**Response to comments of Review 2**

We thank the reviewer for the comments and suggestions on the manuscript. Our response (in blue) and the corresponding edits (in red) are shown below.

**General comments**

This paper presents a model study of the effects of halogen chemistry on the air quality of Europe. This study provides an interesting overview of the impact of halogens on ozone and other pollutants, a research question that is still open. The paper is well written and presented and I have only some minor comments (see below). Overall I think it is suitable for publication in ACP.

1. In Section 3.2 the CMAQ results are compared to the observations and to the GEOSChem results from Sherwen et al. (2017). First of all, there are other observations of ClNO2 in Europe besides those in Table 3. In fact some of these are mentioned in the Sherwen paper itself (as well as in Sommariva et al., 2018) and in Bannan et al. (2017). These measurements should be included in the discussion. Second, using the maximum observed concentration is not a good metric to assess the agreement with the model. For example, the observations in Phillips et al. (2012) show quite a range of peak nocturnal concentrations of ClNO2. I would also argue that GEOS-Chem shows better agreement with the measurements than CMAQ, especially wrt ClNO2 (lines 231-233). The discussion of the model-measurements comparison is better when dealing with iodine and bromine species, but please revise Section 3.2 to be more accurate.

Response: The observational results of $ClNO_2$ reported in Sherwen et al. (2017), Sommariva et al. (2018) and Bannan et al. (2017) have been included in Table 3 of the revised manuscript.

Table 3. The comparison of observed and simulated halogen species

| Location | Species | Observation [*] | Simulation [#] |
|---|---|---|---|
| Hessen, Germany [a] | $ClNO_2$ | 800.0 | 273.4 |
| London, United Kingdom [b] | $ClNO_2$ | 724.0 | 801.5 |
| Weybourne, United Kingdom [c] | $ClNO_2$ | 65 | 373 |
| Weybourne, United Kingdom [d] | $ClNO_2$ | 946 | 373 |
| Weybourne, United Kingdom [e] | $ClNO_2$ | 1100 (summer) 75.6 (autumn) 733 (winter) | 373 |
| Leicester, United Kingdom [e] | $ClNO_2$ | 274 (spring) 74.2 (summer) 248 (winter) | 274 |
| Penlee Point , United Kingdom [e] | $ClNO_2$ | 922 | 319 |

| | | | |
|---|---|---|---|
| Mace Head, Ireland [f] | BrO | 6.5 | 10.1 |
| Brittany, France [g] | BrO | 7.5 | 0.4 |
| Dead Sea [h] | BrO | 100.0 | 0.2 |
| Mace Head, Ireland [i] | IO | 4.0~50.0 | 3.9 |
| Brittany, France [j] | IO | 7.7~30.0 | 1.1 |
| Dagebull, Germany [k] | IO | 2.0 | 9.0 |
| Atlantic Ocean [l] (Prados-Roman et al., 2015) | IO | 0.4 to 0.5 (daytime average) | 0.4 to 2.0 (daytime average) |

\*: Maximum value (pptv).
**: Maximum value (pptv) from the HAL simulation.**
a: Phillips et al. 2012.
b: Bannan et al., 2015.
c: Banna et al., 2017.
d: Sherwen et al., 2017.
e: Sommariva et al., 2018.
f: Saiz-Lopez et al., 2004.
g: Mahajan et al. 2009.
h: Matveev et al., 2001; Holla et al., 2015.
i: Allan et al., 2000; Commane et al., 2011.

Several field campaigns have been conducted in Weybourne in the past few years to measure $ClNO_2$. Sherwen et al. (2017) reported a peak concentration of 946 pptv. Bannan et al. (2019) reported a peak value of 65 pptv, and Sommariva et al. (2018) reported a peak value of 1100 pptv in summer, 75.6 pptv in autumn and 733 pptv in winter. CMAQ simulated a maximum of 373 pptv at that location while GEOS-Chem predicted 458 pptv. Sommariva et al. (2018) also reported measurements of $ClNO_2$ at Leicester with a maximum value of 274 pptv in spring, 74.2 pptv in summer, 248 pptv in winter, and that at Penlee Point a peak value of 922 pptv. CMAQ predicted a maximum of 274 pptv at Leicester and 319 pptv at Penlee Point. Eger et al. (2019) conducted shipborne observation of $ClNO_2$ in the Mediterranean Sea and reported up to 600 pptv $ClNO_2$ during their campaign, which is similar to the prediction of the present study.

We agree with the reviewer that the peak concentration of $ClNO_2$ (normally around the time of sunrise) at one location could have a large range, implying the large day-to-day variation of the level of $ClNO_2$ precursors ($NO_x$, $O_3$, $Cl^-$). The capability of CMAQ model to reproduce the maximum level of $ClNO_2$ at several locations throughout Europe (United Kingdom, Germany, Mediterranean Sea etc.) represents the ability of CMAQ to satisfactorily simulate emission, transport, and chemical transformation processes related to $ClNO_2$. Considering

that the present study was not designed to reproduce the level of $ClNO_2$ in a certain campaign, we think that the current validation metric adequate enough to show that the CMAQ model and the current setting can be used to investigate the halogen impact on the air quality in Europe.

We have removed the sentence that compares the performance of CMAQ with that of GEOS-Chem with regard to $ClNO_2$.

2. Figure 2 is interesting in the sense that it shows some different results from the corresponding figure 5 in the Sherwen paper especially when it comes to BrO. It looks like CMAQ is calculating lower concentrations than GEOS-Chem both for Cl and for HCl, which deserves some comment. It would also be good to include some of the European observations of HCl in this discussion. I realize that a comparison between CMAQ and GEOS-Chem is beyond the scope of this paper, but the differences in the geographical distributions of some species (and related impacts on O3 and other species) are sometimes striking and require at least a brief comment.

Response: CMAQ predicted lower level of BrO compared to GEOS-Chem. We have added a sentence to acknowledge that.

The predicted BrO levels over Europe are low (average value ~0.17 pptv) with the largest predicted value occurring within the Arctic circle while GEOS-Chem predicted >1.0 pptv level of BrO in Mediterranean Sea (Sherwen et al., 2017).

For the modeled level of Cl, CMAQ had a similar distribution and magnitude compared with the GEOS-Chem model, although CMAQ predicted a slightly higher maxima value ($7.0 \times 10^{-4}$ pptv, or $~1.75 \times 10^4$ atom $cm^{-3}$) than the GEOS-Chem model ($1.4 \times 10^4$ atom $cm^{-3}$).

We have also added some discussion on the prediction of HCl level.

The observed level of HCl in Europe is in the range of <100 pptv to 5000 pptv (Hossaini et al., 2016 and the reference therein). The CMAQ model predicted a monthly average concentration of HCl between 6.3 and 1249 pptv, which is similar to the observation range. GEOS-Chem (Sherwen et al., 2017) predicted a maximum of 12 pptv for HCl, which is significantly lower than the available measurements in Europe.

In the original version, we compared the present study with the GEOS-Chem work in the simulation of halogen species ($ClNO_2$, BrO, and IO) oxidants (e.g., OH), and pollutants (e.g. $O_3$) and we also noted the difference between the two models. In the revised version, we have added some more discussions on the difference of the level and distribution of some species of the two models, e.g., BrO and HCl.

3. line 688: correct typo in name.

Response: Corrected. (Dub has been modified to be Dubé)

---

## Author Comment (AC2) · 13 Sep 2019

**Response to comments of Reviewer 1**

We thank the reviewer for the comments and suggestions on the manuscript. Our response (in blue) and the corresponding edits (in red) are shown below.

**General comments**

This work presents the evaluation of the halogen (Cl, Br, I) chemistry scheme from Sarwar et al [2015] within the CMAQ model on a regional scale (12km) for the month of July over Europe. The authors present an evaluation of halogen chemistry's impacts on air-quality metrics by comparing two runs, with chlorine ("CHL") or all halogens (HAL), against a model run without halogen chemistry ("BASE"). The focus is on how halogen chemistry impacts ozone in the European summer.

1. The authors argue the novel aspect of this study derives from the high resolution (12km) and focus on air quality impacts over Europe. However, this novelty is some-what challenged by the extensive referencing of a study in another model that considered also considered halogen chemistry within a nested model over Europe at ~ 25km during the summer too [Sherwen et al 2017]. Furthermore, this work seems to omit reference to the follow-on work using the same nested model ~25km for other seasons [Sommariva et al 2018] or using the model in a global for an entire year at 12km [Hu et al 2018]. It is not clear what the authors are suggesting is the main differences in between the nested global vs. nested hemispheric approaches, apart from the resolution.

Response: We have revised the introduction to describe the novelty of the present study, including (1) using the latest version of the CMAQ model (Sarwar et al., 2019) to investigate the halogen impact on air quality in Europe, (2) the implication for regulation, and (3) the highest spatial resolution used to investigate halogen impact on air quality over Europe.

The revised text in the introduction:

Only a few regional modeling studies have explored the combined influence of the halogen chemistry on air quality. The first modeling study with combined halogen (Cl, Br, and I) chemistry was conducted by Sarwar et al. (2015) who used a hemispheric version of the Community Multiscale Air Quality (CMAQ) model (Ching and Byun, 1999; Byun and

Schere, 2006; Mathur et al., 2017) and reported a decrease of surface $O_3$ by ~15% to ~48% over the Northern Hemisphere by Br and I. Gantt et al. (2017) then utilized the CMAQ model to explore the role of halogen chemistry at a regional scale over the continental United States (US). While these studies focused on Northern Hemisphere and the continental US, Muñiz-Unamunzaga et al. (2018) applied the full-halogen chemistry version of CMAQ with a resolution of 4 km and reported up to 5 ppbv decrease of $O_3$ in the city of Los Angeles, California, US. Sherwen et al. (2017) used a global model, GEOS-Chem, in a regional configuration (with a grid size of 0.25° × 0.315°, ~25km x ~25km) and predicted a large decrease of $O_3$, on average -13.5 pptv (25%) and as much as -28.9 pptv (45%) in Europe. Sarwar et al. (2019) further updated the halogen chemistry in CMAQ model and reported a reduction of -3 to -12 ppbv of annually average $O_3$ over seawater and -3 to -6 ppbv over coastal and -3 ppbv over inland area by Br and I. These previous regional studies using various models (or versions of models) in different areas reported a large range of the halogen impact on $O_3$ highlighting the uncertainty in this research field.

The regulation of air quality and the control of air pollutants emission in Europe started in the early 1970s and over forty years of effort has successfully improved air quality throughout Europe (EEA, 2018a). Nonetheless, poor air quality persist in major cities like Madrid, Paris, and London (EEA, 2018a); this shows the need for continuied air quality management and effective policy. Because the influence of halogens on air quality is uncertain and potentially has an impact on air quality management decisions, we have conducted regional simulations using the latest version of the CMAQ model implemented with comprehensive halogen sources and chemistry (Sarwar et al., 2019) to examine the overall effect of halogen species on air pollution over Europe. Considering that the grid size has a noticeable impact on air quality model predictions (Sommariva et al., 2018), we used a CMAQ model domain with 12 km horizontal resolution (higher than the previous studies on halogen impact covering Europe) to simulate the levels of halogen species over Europe, examine the effect on the oxidation capacity and the concentration of air pollutants, and explore the potential implications for air quality policy related to $NO_2$ and $O_3$.

2. Furthermore when considering the period of study: Halogen chemistry can be very seasonally dependent in Europe (e.g. ClNO2 - See Sommariva et al [2018]). So considering

the existing literature on modelling halogens, further model runs should at least be presented for a winter month to give an equivalent novel value on understanding the air-quality impacts.

Response: The present study focuses on the halogen impact on air quality during summer season. We have revised the title.

Impact of halogen chemistry on summertime air quality in coastal and continental Europe: application of CMAQ model and implication for regulation

3. Some opportunities to make this work more novel seem to have been missed. For instance, coarse two bin comparisons ("coastal" vs. "inland") are made with regional ozone and NO2 observations instead of comparisons that show if the model captures chemical/physical processing (e.g. diel plots by the hour). Oxidants are analysed as "snapshots" of peak or average concentrations, rather than considered in terms of a given hour. Considering the diel cycle of oxidants by hour over Europe (e.g. contribution of Cl in the morning vs. OH at midday, then NO3) and how halogens effect this would really elevate this manuscript.

Response: We have added diurnal plots of $O_3$, $NO_2$, OH, $HO_2$, $NO_3$, and Cl and the corresponding text in the revised manuscript.

[Figure]

Figure S2. Diurnal variation of observation and simulation (BASE and HAL) of $O_3$ and $NO_2$ in northern and southern Europe.

The BASE simulation under-predicts $O_3$ compared to observations both at coastal and continental stations (Table 1), possibly due to the uncertainty of VOC emission inventory (Sherwen et al., 2017) and the underestimated $NO_X$ (Table 1). The HAL simulation slightly improves the correlation coefficient of $O_3$ but decreases the average level of $O_3$ compared to the BASE case. Diurnal variation plots (Fig. S2) suggests that both BASE and HAL simulations produces the temporal patterns of $O_3$ and $NO_2$.

[Figure]

Figure S3. Diurnal variation of simulated (BASE and HAL) OH and $HO_2$ over northern and southern Europe.

[Figure]

Figure S4. Diurnal variation of simulated (BASE and HAL) $NO_3$ and Cl over northern and southern Europe.

We also examine the diurnal variations of the four radicals in the BASE and HAL scenarios (Fig. S3 and S4). Halogens have small effect on the diurnal pattern of OH. $HO_2$ is reduced by halogens especially in the mid-day. $NO_3$ radical is strongly decreased throughout the night after the addition of halogens. Cl atom is released by the halogen chemistry evidently in the

early morning. The significant effects of halogen chemistry on the diurnal variation of OH, $HO_2$, $NO_3$, and Cl radicals highlight the role of halogen chemistry in regulating the atmospheric oxidation capacity throughout the day with the highest effect on Cl in the early morning, maximum effects on OH and $HO_2$ in daytime, and largest effect on $NO_3$ at night.

100

4. The description of the model setup needs to be clearer. From reading the manuscript, I think I correctly understand that the configuration uses a regional grid for Europe and then sets "boundary conditions" outside this grid from two additional hemispheric runs. Please add more information to the model configuration section and clarify this. Writing is often verbose

105 and would often benefit from re-wording to make the manuscript more concise. The introduction especially would benefit for re-writing for flow and clarity of the various different model studies cited. References (e.g. for observations) could be more up to date.

Response: In the revised version of the manuscript, we used the latest version of CMAQ model incorporated with up-to-date halogen chemistry (Sarwar et al., 2019) to re-run all

110 simulation cases.

The boundary condition setting is also modified.

Boundary conditions for the model were derived from the hemispheric CMAQ simulations (Mathur et al., 2017). Three different annual simulations were conducted using the hemispheric CMAQ model for 2016: the first simulation used the Carbon Bond chemical

115 mechanism without any halogen chemistry, the second simulation used the Carbon Bond chemical mechanism and the chlorine chemistry, and the third simulation used the Carbon Bond chemical mechanism and the full halogen chemistry. Results from the corresponding hemispheric CMAQ simulation were used to generate boundary conditions for the BASE, CL, and HAL simulations. Therefore, the difference between CL and BASE simulations

120 represents the impact of the chlorine chemistry on air quality and the difference between HAL and BASE simulations represents the effect of halogen chemistry on air quality.

The spin-up time has also been changed.

The study was completed for the month of July 2016 with a spin-up period of 30 days.

Therefore, all figures, tables, numbers in the text have been modified using the new simulation results. Please note that the conclusions are not changed using the new model, new boundary condition setting, and new spin-up time.

We have also added more reference, e.g. Sommariva et al., (2018) and Bannan et al., (2017), in the halogen simulation (section 3.2).

Table 3. The comparison of observed and simulated halogen species

| Location | Species | Observation [*] | Simulation [#] |
|---|---|---|---|
| Hessen, Germany [a] | $ClNO_2$ | 800.0 | 273.4 |
| London, United Kingdom [b] | $ClNO_2$ | 724.0 | 801.5 |
| Weybourne, United Kingdom [c] | $ClNO_2$ | 65 | 373 |
| Weybourne, United Kingdom [d] | $ClNO_2$ | 946 | 373 |
| Weybourne, United Kingdom [e] | $ClNO_2$ | 1100 (summer) 75.6 (autumn) 733 (winter) | 373 |
| Leicester, United Kingdom [e] | $ClNO_2$ | 274 (spring) 74.2 (summer) 248 (winter) | 274 |
| Penlee Point , United Kingdom [e] | $ClNO_2$ | 922 | 319 |
| Mace Head, Ireland [f] | BrO | 6.5 | 10.1 |
| Brittany, France [g] | BrO | 7.5 | 0.4 |
| Dead Sea [h] | BrO | 100.0 | 0.2 |
| Mace Head, Ireland [i] | IO | 4.0~50.0 | 3.9 |
| Brittany, France [j] | IO | 7.7~30.0 | 1.1 |
| Dagebull, Germany [k] | IO | 2.0 | 9.0 |
| Atlantic Ocean [l] (Prados-Roman et al., 2015) | IO | 0.4 to 0.5 (daytime average) | 0.4 to 2.0 (daytime average) |

[*]: Maximum value (pptv).
[#]: Maximum value (pptv) from the HAL simulation.
a: Phillips et al. 2012.
b: Bannan et al., 2015.
c: Banna et al., 2017.
d: Sherwen et al., 2017.
e: Sommariva et al., 2018.
f: Saiz-Lopez et al., 2004.
g: Mahajan et al. 2009.
h: Matveev et al., 2001; Holla et al., 2015.
i: Allan et al., 2000; Commane et al., 2011.
j: Britter et al., 2005; Furneaux et al., 2010.

k: Peters et al., 2005.

l: Prados-Roman et al., 2015

145 Several field campaigns have been conducted in Weybourne in the past few years to measure $ClNO_2$. Sherwen et al. (2017) reported a peak concentration of 946 pptv. Bannan et al. (2019) reported a peak value of 65 pptv, and Sommariva et al. (2018) reported a peak value of 1100 pptv in summer, 75.6 pptv in autumn and 733 pptv in winter. CMAQ simulated a maximum of 373 pptv at that location while GEOS-Chem predicted 458 pptv. Sommariva et al. (2018)

150 also reported measurements of $ClNO_2$ at Leicester with a maximum value of 274 pptv in spring, 74.2 pptv in summer, 248 pptv in winter, and that at Penlee Point a peak value of 922 pptv. CMAQ predicted a maximum of 274 pptv at Leicester and 319 pptv at Penlee Point. Eger et al. (2019) conducted shipborne observation of $ClNO_2$ in the Mediterranean Sea and reported up to 600 pptv $ClNO_2$ during their campaign, which is similar to the prediction of

155 the present study.

**Specific comments**

5. Page 1 - Lines 15-35 (Abstract) Please add headline numbers for changes seen in this manuscript to abstract.

160 What is the net effect of the domain on OH/HO2? The effect of halogens on OH /HO2 in box models and global models has been discussed at length by Stone et al. [2018].

Response: We have added the predicted values in the abstract.

Combined halogen chemistry induces complex effects on OH (between -0.023 pptv and 0.030 pptv) and $HO_2$ (in the range of -3.7 to 0.73 pptv), significantly reduces the

165 concentrations of $NO_3$ (as much as 20 pptv) and $O_3$ (as much as 10 ppbv), and decreases $NO_2$ in the highly polluted regions (as much as 1.7 ppbv) and increases $NO_2$ (up to 0.20 ppbv) in other areas.

The net effect in the domain of halogens on OH and HO2 is added in the revised manuscript.

The net change of OH due to halogen was near zero.

170 The overall difference of $HO_2$ because of halogens was -0.59 pptv in the European domain.

The reference of Stone et al. (2018) is added in the discussion of the halogen impact on $HO_x$.

Another GEOS-Chem study, however, predicted an increase of OH over the Mediterranean Sea (Stone et al., 2018).

175    6. Page 2 - Line 35 - 41 Earlier references should be used here to give fair credit to the original work on this, instead of recent reviews.

It would be cleaner and more instructive to the reader to include the following sentence, rather than just citing a recent review.

"The chemistry of halogens in the troposphere has been described in detail in recent reviews
180    (Saiz-Lopez and von Glasow, 2012; Simpson et al., 2015), so we just briefly outline it here."

Additionally, the effect of halogen nitrate hydrolysis on decreasing ozone production should be mentioned here as it has been shown to have a larger impact on ozone than increased loss [Schmidt et al 2016].

Response: We have added the suggested sentence in the introduction.

185    The chemistry of halogens in the troposphere has been described in detail in recent reviews and references therein (Saiz-Lopez and von Glasow, 2012; Simpson et al., 2015), so it is just briefly outlined here.

We have also added the effect of the hydrolysis of halogen nitrate on the $O_3$ level in the introduction.

190    indirectly decreasing $O_3$ production by reducing $NO_2$ (R2 and R3),

$$XO + NO_2 \rightarrow XONO_2 \tag{R2}$$

$$XONO_2 + H_2O \text{ (l)} \rightarrow HOBr + HNO_3\text{(l)} \tag{R3}$$

7. Page 2 - Line 46 "methane ch4" should read "methane (CH4)"

195    Response: Revised.

8. Page 2 - Line 52-55 "Evaluation of the complex role of halogen chemistry in air quality requires the employment of advanced, high-resolution chemical transport models"

Has this argument really been evidenced here? Could not the processed discussed be captured by existing coarse resolution approaches? Are the uncertainties on halogen modelling substantially small enough that horizontal resolution of models is the main limitation for the fidelity of simulation to observations and capturing chemical and physical processes?

Response: Please refer to response 1.

9. Page 2 Line 56-61 Should not the faraday discussions paper you mentioned earlier be included here as you say in the manuscript it covered ClNO2 too? Other model studies have looked at this too and should be included here also. Adding "e.g. X et al., Y et al." to illustrate the reader that a couple of examples have been given would also be appropriate. There are other points in the manuscript where this would be appropriate too.

Response: Added more reference to the previous studies on simulating $ClNO_2$, including Sherwen et al. (2017).

The chemistry of chlorine, mainly that of $ClNO_2$, has been reported to increase the oxidation capacity and the formation of $O_3$ in recent studies (Sarwar et al., 2012, 2014; Li et al., 2016; Sherwen et al., 2017; Sommariva et al., 2018).

10. Page 3 - 61-71 It is hard for the reader to follow the way in which the previous work is being discussed. The authors have referred to another nested regional study in GEOSChem in the final intro paragraph [Sherwen et al 2017], but then did not include it in the discussion of existing regional modelling work here.

Response: Added Sherwen et al. (2017) in the description of regional model simulation.

Sherwen et al. (2017) used a global model, GEOS-Chem, in a regional configuration (with a grid size of 0.25° × 0.315°, ~25km × ~25km) and predicted a large decrease of $O_3$, on average 13.5 pptv (25%) and as much as 28.9 pptv (45%) in Europe.

11. Page 3 - Line 71 It is arguable that 4km is a higher resolution. However, is a 4km horizontal resolution accepted to be sufficient to represent the processes going on in a city? I would suggest updating as follows:

From "halogen sources on air quality at a city scale (4 km resolution) in Los Angeles, California, US." To "halogen sources on air quality at a resolution of 4 km in the city of Los Angeles (California, US). "

Response: Revised.

12. Page 3 - Line 74 A URL in brackets is not an appropriate format here. If a URL must be used please include a reference to the EEA and a data accessed for the data. Referencing an EEA report would be preferable.

Response: Revised.

13. Page 3 - Line 72 to 82 It is hard for the reader to follow the semantic. The previous study is also a domain based study over Europe using boundary conditions from a model with a larger (global) domain with offline meteorology. Are the authors arguing that the main difference between the 12x12km CMAQ approach presented here and the existing work, at the coarser resolution (25x25km), is just resolution? Both models use similar halogen chemistry are nested within larger domains, correct? Why is so much change in simulation skill expected to be seen between 12x12 km and 25x25km? Or would a higher resolution, say 4x4 km [MunÌCˇ iz-Unamunzaga et al., 2018], be required to notable gains in capturing processes or differences? This leads to a more philosophical question: is it a model resolution or processes holding back science currently? Would not other uncertainties in halogen chemistry be greater than the difference caused by a change in resolution too? (e.g. emissions developed for coarser resolutions or new developments in the representation of halogen chemistry in models - Xuan et al [2019])

Response: Please refer to response 1.

14. Page 3 - Line 84 Is "instrumented" the appropriate word here? "Including" would be a better choice or "which includes".

Response: "instrumented" has been changed to "implemented".

15. Page 3 - Line 85 Why is "state-of-the-art" used here? It seems unnecessarily verbose, especially when referring to a paper that is at least four years old in a fast-moving part of the literature.

Response: In the revised version of manuscript, we use the updated halogen chemistry in CMAQ (Sarwar et al., 2019). We have revised the paragraph as follows:

The regulation of air quality and the control of air pollutants emission in Europe started in the early 1970s and over forty years of effort has successfully improved air quality throughout Europe (EEA, 2018a). Nonetheless, poor air quality persist in major cities like Madrid, Paris, and London (EEA, 2018a); this shows the need for continuied air quality management and effective policy. Because the influence of halogens on air quality is uncertain and potentially has an impact on air quality management decisions, we have conducted regional simulations using the latest version of the CMAQ model implemented with comprehensive halogen sources and chemistry (Sarwar et al., 2019) to examine the overall effect of halogen species on air pollution over Europe. Considering that the grid size has a noticeable impact on air quality model predictions (Sommariva et al., 2018), we used a CMAQ model domain with 12 km horizontal resolution (higher than the previous studies on halogen impact covering Europe) to simulate the levels of halogen species over Europe, examine the effect on the oxidation capacity and the concentration of air pollutants, and explore the potential implications for air quality policy related to $NO_2$ and $O_3$.

16. Page 4 - Line 90-96 The meteorology is offline? Or is CMAQ being run in coupled mode? Please explicitly state if the meteorology is offline.

Response: The CMAQ model was run offline.

280  The meteorological inputs for the CMAQ model were obtained from the Weather Research and Forecasting model (WRF 3.7.1) (Skamarock and Klemp, 2008; Borge et al., 2008a)  as an offline input.

17. Page 4 - Line 96 Why is the word "besides" used in this sentence. Please rephrase for
285  clarity.

Response: Removed.

18. Page 4 - Line 111 How has coastal been defined here? More broadly, is 12km sufficient resolution to capture "coastal" effects?

290  Response: Previous studies (Sarwar et al., 2015; Sherwen et al., 2016; Sarwar et al., 2019) suggest that halogen chemistry affects $O_3$ not only over marine environments but also over inland locations far away from marine environments. Thus, we use coastal area to include greater land area adjacent to marine environments and employ all monitoring stations within 24-km from the coastline to examine the impact of the halogen chemistry impact on coastal
295  areas as well as over inland areas.

19. Page 6 - Line 131 -132 Is this 12x12km domain nested within another domain not computed online? Or is the model run hemispherically at 12x12km here for all simulations? Please make this clearer.

300  Response: In the present study, the CMAQ model is run with one domain with a spatial resolution of 12x12 km. The hemispherical simulation is only used as the boundary condition.

The CMAQ model is  applied over a domain covering the entirety of Europe (Fig. 1) with 12 km horizontal resolution.

305

20. Page 6 - line 143 The name "CHL" may make many readers from the Earth sciences community think of chlorophyll. I would suggest using "Cl" instead to make the paper more accessible to a broader readership.

Response: Revised.

310

21. Page 6 - Line 139 -145 Only 7 days spin up was used for a hemispheric simulation? Or is this for the European grid? Is this sufficient to ensure initial conditions are "washed" through the grid? Is there a reference showing this is sufficient? What spin up was used for "the hemispheric CMAQ simulations"/hemispheric grid? What initial conditions were used? This needs to be clearer. Global studies on halogens in CMAQ, GEOS-chem and CAM-Chem have highlighted the importance of changes in background concentrations. The relative contribution of boundary and local effects seems to be a core focus of the Sherwen et al [2017] manuscript, but not really given much attention or discussion here.

Response: The hemispheric simulation of CMAQ, which is used for the boundary condition, is described in Mathur et al. (2017).

In the revised version, we have used a spin-up period of 30 days.

In the present study, we focus on the overall impact of halogens on air quality in Europe, instead of the relative contribution of boundary and local effects.

325

22. Page 6 - Line 152-160 The model is being run on a hemispheric grid of 12x12, but just analysed for the domain in Figure 1? The "Boundary conditions" are on the same grid, but global? and these are provided to the outside of the modelling domain? This needs to be explained more clearly.

Response: Please refer to response 19 and 21.

330

23. Page 6 - Line 152 "Boundary conditions for the model were derived from the hemispheric CMAQ simulations."

Which hemispheric CMAQ simulations? Where have these been described?

Response: Please refer to response 21.

335

24. Page 7 - Line 161-163 Please include a table that shows which boundary conditions and chemistry used in each domain for each model experiment.

"the difference between CHL and BASE simulations represents the impact of the chlorine chemistry on air quality"

340 Don't both of these simulations have the same boundary conditions ("Carbon Bond chemical mechanism and the chlorine chemistry")? Surely then the most this difference can show is the contribution of chlorine chemistry locally. A preferable approach would be to use a "BASE" set of boundary conditions without halogen chemistry for the "BASE" simulation.

"the difference between HAL and BASE simulations represents the effect of halogen
345 chemistry on air quality."

Again: according to the text, "BASE" includes the effects of chlorine globally as it includes chlorine in boundary conditions ("Carbon Bond chemical mechanism and the chlorine chemistry"). Therefore "BASE" - "HAL" is giving the effects of halogens minus the global effect of chlorine.

350 Although locally chlorine can provide an oxidant effect and lead to ozone formation, it also can act as a sink for ozone through the loss of chlorine nitrates on a global scale. This has been discussed in some of the global modelling papers cited here. What are the global effects of chlorine in this model? Can global effects be excluded here?

Response: Please refer to response 4.

355

25. Page 7 - line 169 Please give a justification for the use for "within 24 km from the coast" definition. Coastal processes (notably halogens) can be confined to a very small area (e.g. macroalgae in tidal zones).

Response: Please refer to response 18.

360

26. Page 7 - Line 169-171 The level of the evaluation presented needs to be increased to be in line with the high-resolution output the authors argue is notable here.

Please plot ozone and NO2 as a diel (24-hour) comparison compared with the model, preferably showing a few regions in Europe. Regions like the coastal Mediterranean should not just be lumped together with the coastal Scandinavian, as they have very different seasonal characteristics. Simply using a table to do a very coarse two bin comparison does not provide much insight and it is not really in line with the current level analysis presented in the literature (e.g. Schnell et al. 2015), instead, it smooths out the extra information gained.

Response: Please refer to response 3.

27. Page 7 - Line 179-181 Please provide references for the proposed explanations for model bias or expand this discussion.

Response: We have added one reference, Jung et al., 2017, for the explanation of the $NO_2$ simulation.

Jung, J., Lee, J., Kim, B. and Oh, S.: Seasonal variations in the NO2 artifact from chemiluminescence measurements with a molybdenum converter at a suburban site in Korea (downwind of the Asian continental outflow) during 2015–2016. Atmos. Environ., 165, 290-300, 2017.

28. Page 7 - Line 184-186 Only a single table of comparisons has been provided to back up this statement. More evidence is needed. Please provide diel plots of core species (e.g. CO, NO2, O3).

Response: Please refer to response 3.

29. Page 7 - Line 182 How does this comparison look on a diel basis? Is there an offset or difference in the diel cycle at certain times of the day? Does the model capture the diel cycle before or after or is there a structural issue in the model (e.g. caused by emissions or boundary layer mixing?)?

Response: Please refer to response 3.

390

30. Page 11 - Line 223-255 The coastal influence or inland nature of observations should be made clearer. At the resolution presented here (12x12km) it would not be fair to expect the model to reproduce many of these observations (e.g those influenced by local emissions from the tidal zone).

395    Response: Please refer to response 18.

31. Page 11 - Table 1 This table of observations does not seem in line with current literature (e.g. observations for ClNO2 are available across seasons in Northern Europe [Sommeriva et al 2018]). Please expand these comparisons.

400    Response: Please refer to response 4.

32. Page 11 - Line 249 The largest IO dataset has been published since Saiz-Lopez and von Glasow [2012] by Prados-Roman [2015]. Did this not this start and end in the Mediterranean? How does the model compare against this? What other more recent datasets

405    are there?

Response: We have added the comparison between the CMAQ modeling results with the observations reported in Prados-Roman et al. (2015).

Prados-Roman et al. (2015) reported the level of IO during a ship-based campaign in the range of <0.4 to >1.4 pptv (daytime average) around the globe and 0.4 to 0.5 pptv

410    (daytime average) in the south of Spain and the west of Africa (over the Atlantic), and the present study predicted 0.4 to 2.0 pptv (daytime average) of IO in that area.

33. Page 12 - Line 261-264 Please see earlier comments about the inclusion of chlorine in boundary conditions for both "BASE" and "CHL" simulations.

415    Response: Please refer to response 4.

34. Page 13 - Line 267 Please give OH units in the more commonly used units of molecules cm-3 or at least show this in brackets throughout the text. Please do the same for Cl (atoms cm-3).

Response: Revised.

35. Page 13 - Line 290 What about the resolution difference could cause this? What other differences could explain this? Are there any differences in the chemistry between the Sarwar et al (2015) and Sherwen et al (2017) Mechanism?

Response: We have revised the discussion.

Another GEOS-Chem study, however, predicted an increase of OH over the Mediterranean Sea (Stone et al., 2018). The discrepancy among the previous studies and between those works and the present one is difficult to deduce and requires further investigation. Several possible causes could lead to different simulated levels of halogens and their impact on oxidants, including the different mechanism of producing and recycling the halogen species (Sarwar et al., 2019), spatial resolution (Sommariva et al., 2018), emission inventory (Wang et al., 2019), and different spatio-temporal scale of interest (Stone et al., 2018).

36. Page 15 - Line 305 "Our study, along with the previous work, highlights the vital role of halogen chemistry in the nighttime chemistry." Which previous work? How is this chemistry constrained by lab work/observations?

Is it based on theoretical calculations? If So, then the uncertainty on this should be highlighted here.

Response: The sentence has been removed.

37. page 15 Line 307 - 309 Cly is in the boundary conditions for both "BASE" and "CHL", correct? How much Cly is transported into the domain?

Response: Please refer to the response 4.

38. Page 15 - 312-318 How do these values compare against known constraints on tropospheric chlorine (e.g. Gromov et al [2018])? Are there any explanations for the differences? (e.g. It is worth noting too that Hossani et al [2016] used unrealistic anthropogenic chloride emissions - see Xuan et al [2019] for details on this)

Response: The suggested reference, Gromov et al., 2018, is a study on the role of chlorine on methane in the southern hemisphere, which is not relevant to the present study. Their reported range of the concentration of Cl, $9 \times 10^3$ to $2.8 \times 10^4$ atom cm$^{-3}$, is within the range in the review we cited (Saiz-Lopez and von Glasow, 2012), $10^3$ to $10^5$ atom cm$^{-3}$.

We have added the following sentence to acknowledge the work by Wang et al. (2019).

In their study, Hossaini et al. (2016) used the Reactive Emission Inventory of Chlorine (Keene et al., 1999) which Wang et al. (2019) reported to be unrealistic for present day applications.

39. page 15 Line 320 "The current study and the previous works simulated a broad range of the surface Cl concentrations although they were all within the scope of the reported observed (observation- based calculation) values of 103 to 105 atom cm-3 (4.0 x10-5 to 4.0 x10-3 pptv) according to the review of Saiz-Lopez and von Glasow (2012)."

Why is a review that is > seven years old being used as the basis for comparison? There have been a large amount of Cly measurements since then (e.g. Gromov et al [2018], Haskins et al [2018] etc ...) and more work to constrain tropospheric Cl [Gromov et al 2018].

Response:   For the suggested reference, Gromov et al. (2018), please refer to the response 38.

The other reference, Haskins et al. (2018), is a study on the measurements of inorganic chlorine species in US. They focused on the partitioning of the gaseous and particulate chlorine and provided constraints on the total atmospheric inorganic chlorine, not the chlorine atom.

40. page 22 Fig 8 AOT40 is calculated over a growing season. Why is this shown for a single month in Fig. 8 and discussed in the text? It would be more appropriate to give output in units of exceedances for a given month as this usage of "AOT40" could mislead readers. What over relevant ozone thresholds are there? What about the particulate matter if the oxidants have changed (e.g. PM10, PM2.5)?

Response: We have revised the manuscript. We used the threshold of 40 ppbv as in AOT40.

Here we use the simulation results (BASE and HAL) in July to calculate the accumulated O$_3$ and the difference between two scenarios. We note that the accumulated O$_3$ is noticeably reduced (>15% along the coast) after the addition of halogens.

[Figure]

Figure 8. Accumulated O$_3$ in July in the BASE and HAL simulations, and absolute and relative changes between the two simulations."

We investigated the halogen impact on oxidants, O$_3$, and NO$_2$ in the present study. The impact on aerosols is not the focus.

41. page 23 Lines 430-432 "These models do not include the comprehensive halogen chemistry, potentially leading to [an] unrealistic simulation of O3 concentration responsiveness to the predicted NOx and/or VOCs emission changes in Europe."

As only a coarse comparison is provided for ozone (table 1) and no diel cycles are shown it is hard to see if halogens are aiding the capture of processes seen within the observations. This weakens the argument that halogen processes are needed to gain a "realistic" simulation in air quality models. Please back up this claim with figures.

Response: Please refer to response 3. We have also modified this sentence:

The models which do not include the comprehensive halogen chemistry can potentially lead to different $O_3$ concentration responses to $NO_x$ and/or VOC emission changes in Europe.

42. page 24 Lines 460-465 The final paragraph of the conclusion comes across as vague. Please provide a few examples of uncertainties that are more specific than just "chemical mechanism".

Response: We have added a few examples of uncertainties in the final paragraph.

Although the incorporation of the halogen chemistry may improve the capabilities of 3D Eulerian chemical transport models, we acknowledge that large uncertainties still exist in the assessment of halogen chemistry impact due to emission inventories (e.g., chlorine emission inventory; Wang et al., 2019), model configuration (e.g., grid size; Sommariva et al., 2018), chemical mechanism (e.g., photolysis rate of iodine oxides, recycling rate of halogen species on aerosol; Simpson et al., 2015), etc.

---

## Author Response (AR2)

**Response to comments of Reviewer 1**

We thank the reviewer for the comments and suggestions on the manuscript. Our response (in blue) and the corresponding edits (in red) are shown below.

1. The authors have responded to comments raised, updated the model version they use, and run additional simulations.

I only have one comment I wish to re-raise from the original review, which is that I remain concerned by the use of AOT40 in the text, when only a single month is modelled. This metric explicitly considers accumulated ozone over the whole growing season. A longer simulation needs to be run or a phrasing germane to "threshold of 40 ppbv" or "Σ(hours>40ppbv)" should be used instead of AOT40 in text and figures.

Response: We have modified our section "3.5. Implications for policy assessment" to address the reviewer concern. It is true that a 3-month simulation would be needed to cover the reference period used for this metric (May to July, for the protection of crops). The AOT40 concept however, can be applied to other accumulation period such us April to September, in the case of forests. This was already reflected in previous versions of our manuscript but we now clarify the limitations of the assessment regarding this legal parameter.

The revised text:

Since our experiment covers only one month, it is not possible to assess the impact of halogen chemistry on these two indexes. Nonetheless, we have compared the results of both maximum daily 8-hour mean and 1-month AOT (Σ(hours>40ppbv) over July; Fig. 8) for the BASE and HAL simulations. The relative variation provide a good indication of the impact that considering halogens in our modelling system may have in the estimation of legally-relevant indexes.

Figure 8. AOT40 for July in the BASE and HAL simulations, and absolute and relative changes between the two simulations

2. I have also raised some minor comments below. After this point have been addressed, I would support publication in ACP.

Fig S2 - Add a definition of Northern and Southern Europe in caption/text (e.g. latitudinal divide used).

We have added the definition of northern and southern Europe in the caption.

3. Table 3 footer - correct typo in the name ("Banna" should be "Bannan").

We have corrected the typo.

4. Page 12 Line 251 - correct typo in reference. "Bannan et al. (2019)" should be "Bannan et al. (2017)"

We have corrected the typo.